# Evaluating biogeophysical sensitivities to idealized deforestation in CMIP6 models using observational constraints

Nikolina Mileva<sup>1</sup>, Julia Pongratz<sup>2,3</sup>, Vivek K. Arora<sup>4</sup>, Akihiko Ito<sup>5,6,7</sup>, Sebastiaan Luyssaert<sup>8</sup>, Sonali S. McDermid<sup>9,10</sup>, Paul A. Miller<sup>11</sup>, Daniele Peano<sup>12</sup>, Roland Séférian<sup>13</sup>, Yanwu Zhang<sup>14</sup>, and Wolfgang Buermann<sup>1</sup>

**Correspondence:** Nikolina Mileva (nikolina mileva@yahoo.com)

Abstract. Forests are an important component in the framework of nature-based solutions for mitigating climate change. However, there are still uncertainties about the biogeophysical effects of forest cover changes affecting heat and water fluxes as captured by Earth System Models (ESMs) simulations and observations. In this study, we investigate the differences in the surface temperature response to idealized, complete deforestation and the temperature sensitivity to percentage change in forest cover in ESMs and observations. In this comparison, the separation between local (at the place of deforestation) and non-local (nearby or distant locations) effects is crucial as observations capture only the former. Here, we propose a modified methodology to separate local and non-local effects in climate models suitable for simulations with linear rate of deforestation. The local sensitivity of a climate variable per unit deforested area is represented by the slope of the linear regression, where tree cover is an explanatory variable. The non-local effect is defined as the difference between the overall change in the respective climate variable and the local effect. Our analysis of eleven ESMs of the Coupled Model Intercomparison Project Phase 6 (CMIP6) that participated in the idealized global deforestation experiment *deforest-glob*, reveals a coherent local temperature response among climate models characterized by warming in the tropics and cooling in the northern higher latitudes. The temperature response however varies in magnitude, space and time with ESMs showing distinctive seasonal and spatial patterns. A closer look at the albedo response to deforestation across norther-northern latitudes shows an overestimation in the ESMs in com-

<sup>&</sup>lt;sup>1</sup>Institute of Geography, University of Augsburg, Augsburg, Germany

<sup>&</sup>lt;sup>2</sup>Department of Geography, Ludwig-Maximilians-Universität München, Munich, Germany

<sup>&</sup>lt;sup>3</sup>Max Planck Institute for Meteorology, Hamburg, Germany

<sup>&</sup>lt;sup>4</sup>Canadian Centre for Climate Modelling and Analysis, Climate Research Division, Environment Canada, Victoria, BC, Canada

<sup>&</sup>lt;sup>5</sup>Graduate School of Life and Agricultural Sciences, The University of Tokyo, Tokyo, 113-8657, Japan

<sup>&</sup>lt;sup>6</sup>Japan Agency for Marine-Earth Science and Technology (JAMSTEC), Yokohama, 237-0061, Japan

<sup>&</sup>lt;sup>7</sup>National Institute for Environmental Studies (NIES), Tsukuba, Ibaraki, 305-8506, Japan

<sup>&</sup>lt;sup>8</sup>Department of Ecological Sciences, Faculty of Sciences, Vrije Universiteit Amsterdam, Amsterdam 1081 HV, the Netherlands

<sup>&</sup>lt;sup>9</sup>Department of Environmental Studies, New York University, New York, NY, USA

<sup>&</sup>lt;sup>10</sup>NASA Goddard Institute for Space Studies, New York, NY, USA

<sup>&</sup>lt;sup>11</sup>Department of Physical Geography and Ecosystem Science, Lund University, Sweden

<sup>&</sup>lt;sup>12</sup>CMCC Foundation - Euro-Mediterranean Center on Climate Change, Bologna, Italy.

<sup>&</sup>lt;sup>13</sup>CNRM (Université de Toulouse, Météo-France, CNRS), Toulouse, France

<sup>&</sup>lt;sup>14</sup>CMA Earth System Modeling and Prediction Centre, China Meteorological Administration, Beijing, China

parison to observations that translates via an emergent constraint (i.e. resulting from the linear relationships within the model ensemble) into an overestimation of the overall simulated cooling effect. The overestimation of the local albedo sensitivity cannot be explained solely by the higher percentage of snow cover in ESMs. In terms of local latent heat flux sensitivity, the ESMs ensemble mean is overestimated for the boreal region, but it is in good agreement with the observational constraint in the temperate forests and the tropics. However, the inter-model spread and the internal model variation in these regions are considerable. ESMs having higher local albedo and latent heat flux sensitivities than the current observational constraints can still exhibit a realistic temperature response due to compensatory effects between the two sensitivities. Non-local effects contribute to consistent cooling throughout the globe, which persists also during the summer when the influence of the overestimated albedo sensitivity over snow is weaker. Having a deeper understanding of how local and non-local biogeophysical effects are represented in ESMs can give us insights into the net climate impact of deforestation and help us improve next generation ESMs.

Copyright statement. TEXT

#### 1 Introduction

Forests are essential for our adaptation to a warmer world as they cool the climate during the hottest months of the year, contributing to the resilience of urban, agricultural, and natural landscapes (Lawrence et al., 2022). Through their climate benefits, forests can also serve as one of the most promising natural climate solutions - a set of measures aimed at mitigating climate change without limiting the supply of food and fiber and putting natural habitats under pressure (Griscom et al., 2017). Whether through afforestation, reforestation, avoided forest conversion, improved forest management, forest restoration or agroforestry, forests have a large potential for capturing and retaining CO<sub>2</sub> (Griscom et al., 2017). In addition to acting as a carbon sink, forests also influence the climate by altering key biogeophysical properties such as albedo, evapotranspiration efficiency and surface roughness (Bonan, 2008). Unlike the biogeochemical effects of deforestation, which encompass an increase in CO<sub>2</sub> concentration in the atmosphere with a clear link to warming, the biogeophysical effects of changes in forest cover can have opposite impacts on temperature depending on the location and season (Bonan, 2008) and the type of forest in question (Bright et al., 2017; Naudts et al., 2016). For example, boreal forests can have a warming effect (relative to nonforested boreal regions) because of their significantly lower albedo compared to snow-covered short vegetation, while tropical forests can cool the climate through higher rates of evapotranspiration and enhanced cloud cover (Betts, 2000; Claussen et al., 2001; Wang et al., 2009). Correspondingly, deforestation has a cooling effect in the boreal region driven mainly by the higher albedo due to the reduction in the snow-masking effect of trees, thus limiting the amount of available energy at the surface. In tropical forests, the albedo effect is not as strong and is overpowered by the increase in incoming shortwave radiation due to lower cloud cover. The increase in incoming solar radiation leads to a rise in net surface radiation and consequently of surface temperature as evaporative cooling over grasslands is not as efficient (Boysen et al., 2020).

To gain more insight into the biogeophysical effects on climate resulting from large-scale changes in forest cover, a number of Earth System Models (ESMs) and regional climate models experiments have been performed using either plausible deforestation patterns derived from historic or future land use and land cover changes (e.g., Pongratz et al., 2010; Boisier et al., 2012; Lejeune et al., 2018; Li et al., 2023), or idealized extensive deforestation scenarios allowing stronger signal-to-noise ratio ((e.g., Durbidge et al., 1993; Werth and Avissar, 2002) and more recently (Devaraju et al., 2018; Strandberg and Kjellström, 2019; Boysen et al., 2020)). Previous deforestation studies (e.g., Bala et al., 2007; Davin and de Noblet-Ducoudré, 2010; Bright et al., 2017; Winckler et al., 2019b) have identified the competing effects of decreasing evapotranspiration efficiency and surface roughness, which typically lead to warming, and increasing albedo, which is considered to be the dominant of the three factors driving global mean cooling (Davin and de Noblet-Ducoudré, 2010; Laguë et al., 2019). In such model simulations, deforestation causes strong changes in surface temperatures locally (at the place of deforestation) through changes in biogeophysical land surface properties (termed "the local effect"). But large-scale deforestation in these experiments also triggers strong changes in advection of heat and moisture, as well as in atmospheric and ocean circulation, which influence regions that have not undergone deforestation (Winckler et al., 2017) (Winckler et al., 2017; Portmann et al., 2022). These biogeophysical "non-local effects" may be even more important than local effects in terms of their influence on the patterns of surface temperatures (Winckler et al., 2019a) temperatures (Winckler et al., 2019a, c). By considering only local effects in models, the global mean cooling observed in complete deforestation experiments could be to a large extent reconciled with the warming pattern in observations (Winckler et al., 2019a; Chen and Dirmeyer, 2020).

More recently, substantial strides in our understanding of the impact of changes in forest cover on near-surface climate could be achieved also from an observational perspective through the availability of high-quality satellite-based data products of key climate variables (e.g., surface temperature) and tree cover (e.g., Alkama and Cescatti, 2016). Similarly to models, observation-based studies recognized the competing effects of evapotranspiration and albedo in forests (e.g., Li et al., 2015). Deforestation in the arid, tropical and temperate regions leads to an increase in the mean surface and air surface temperature (Alkama and Cescatti, 2016; Bright et al., 2017; Duveiller et al., 2018b). In the boreal region, a clear seasonal pattern is observed showing warming during the snow free months and cooling during the rest of the year with mean annual effect ranging from mild warming to significant cooling (Alkama and Cescatti, 2016; Li et al., 2016). In these studies, the climate impact of deforestation is usually estimated between neighboring pixels with contrasting forest cover - the so called space-fortime substitution, using the difference in temperature before and after deforestation (e.g., Alkama and Cescatti, 2016; Li et al., 2016; Baker and Spracklen, 2019; Prevedello et al., 2019), or between climatology averages (e.g., Li et al., 2015; Duveiller et al., 2018b) to account for natural climate variability. Importantly, both approaches only capture the local biogeophysical effects of deforestation as non-local effects are either canceled out or indistinguishable from natural climate variability. Another limitation of satellite observations is that they are collected during mostly cloud-free days, which can bias estimates of surface temperature changes resulting from changes in forest cover (Chen and Dirmeyer, 2020). A new method proposed by Bright et al. (2017) overcomes this limitation by deriving empirical estimates of the local surface temperature change using flux tower measurements, which are collected continuously also during overcast conditions.

Comparing only the local biogeophysical effects of deforestation in ESMs and observations has reconciled many of the earlier discrepancies in the findings based on these two approaches, particularly for northern latitudes (Pongratz et al., 2021). Yet, substantial differences in the magnitude of the local temperature response remain in models and satellite-based studies, especially in certain areas such as the boreal region and southern tropics (Winckler et al., 2019a). In addition, when considering the combined (local and non-local) effects of deforestation, ESMs show substantial differences in temperature sensitivities to deforestation, sometimes even with opposite sign (Boysen et al., 2020). Land surface models, an integral part of ESMs, still have limitations in simulating turbulent heat fluxes, leading to discrepancies and even disagreement between models in the sign of change triggered by land cover transitions (Duveiller et al., 2018a). Uncertainties exist also in the satellite-based temperature sensitivities, which can come from differences in the resolution and sensor accuracy or the underlying parametrization (Chen and Dirmeyer, 2020).

One of the key goals of this study is to provide observational-based emergent constraints (Hall and Qu, 2006) for the ESM responses to deforestation by comparing local surface temperature and key biogeophysical sensitivities to changes in forest cover in observations and models. Observations cannot be directly compared against ESMs because the spatial extent, location and background climate conditions determine the biogeophysical response to deforestation and differ between the observations and simulations. However, local sensitivities are largely independent of these effects, allowing us to apply the emergent constraints approach. The emergent constraints concept states that a relationship between two variables can emerge across simulations with different background climate (i.e. current climate and future climate projection) in a sufficiently large ensemble of ESMs. By knowing the local effects from observations and the emerging relationship between local and total effects from model experiments, we could use these relationships to constrain the total effects of deforestation on the near-surface climate.

In this study, we address therefore specifically the question whether the local effects of deforestation are consistent across a range of ESMs and how well these agree with observation-based observed in-situ and satellite-based local responses. We investigate the differences in the local temperature response to deforestation (i.e. temperature sensitivity) in ESMs and observations and relate them to key biogeophysical properties controlling the interactions between forests and near-surface climate. The objective is to provide observationally based emergent constraints for local surface temperature, albedo and latent heat flux sensitivities to deforestation, against which the ESM based sensitivities can be compared. This evaluation of the simulated local climate effects of large-scale deforestation is a first step toward more robust simulations of the total (local and non-local) biogeophysical climate impact of large-scale afforestation and reforestation efforts.

## 2 Methods




## 2.1 ESM deforestation experiment

Idealized deforestation experiments have a higher signal-to-noise ratio compared to realistic deforestation scenarios, which allows the deforestation signal to exceed model internal variability (Davin et al., 2010). The Land Use Model Intercomparison Project (LUMIP), endorsed by CMIP6, provides a set of experiments aiming to quantify the effects of land use and land cover

change on climate (Lawrence et al., 2016). In this study, we focus on the idealized global deforestation experiment *deforest-glob* conducted as part of LUMIP. In this simulation, roughly 20 million km² of forest are converted to natural grassland over a period of 50 years, followed by 30 years with stable forest cover. The deforestation is performed in grid cells having the highest percentage of tree cover area, thus creating a similar pattern of deforestation across the ESMs, limited mostly to the boreal and tropical regions—(Fig. 1). The climate and anthropogenic forcings are kept at pre-industrial level by branching off the deforestation experiment from an 1850 control simulation (*piControl*) as defined by CMIP (Eyring et al., 2016). Dynamic vegetation changes within deforested areas are disabled to prevent the regrowth of trees. A more detailed description of the simulations is available in Lawrence et al. (2016). The ESMs that provided results for the *deforest-glob* experiment included: MPI-ESM-1.2.0 (MPI) (Wieners et al., 2019; Pongratz et al., 2019), IPSL-CM6A-LR (IPSL) (Boucher et al., 2018, 2019), CESM2 (CESM) (Danabasoglu et al., 2019; Danabasoglu, 2019), CanESM5 (CanESM) (Swart et al., 2019b, a), CNRM-ESM2-1 (CNRM) (Séférian, 2018, 2019), BCC-CSM2-MR (BCC) (Wu et al., 2018; Zhang et al., 2019), MIROC-ES2L (MIROC) (Hajima et al., 2019; Ito and Hajima, 2020), UKESM1-0-LL (UKESM) (Tang et al., 2019; Wiltshire, 2020), EC-Earth3-Veg (EC-Earth) (Döscher et al., 2022; EC-Earth Consortium, 2019, 2020), CMCC-ESM2 (CMCC) (Lovato et al., 2021; Peano et al., 2021), and GISS-E2-1-G (GISS) (NASA Goddard Institute for Space Studies, 2018, 2020). All models couple land, atmosphere and ocean in terms of momentum, matter and energy (Lawrence et al., 2016).

The ESM simulations provide all required variables – surface temperature, albedo, latent heat flux, snow cover and forest cover changes – from the same simulations. Because the albedos <u>ealculated\_retrieved</u> by the ESMs were not available from the model output, the albedo used in this study was calculated as the ratio between surface upwelling shortwave radiation and incoming shortwave radiation. The corresponding names conforming to the climate and forecast model conventions and CMIP standards are provided in Table A1. All analyses were performed at the original grid resolution except for Fig. 69, where the models' outputs were resampled to a common 1.25°x0.94° grid using the nearest neighbor algorithm.

#### 2.2 Observational datasets






Bright et al. (2017) combine remote sensing and in situ in situ measurements (from FLUXNET) to derive the local surface temperature response to different land use and land cover changes. In their dataset, nine common land cover and land management transitions are studied. Here, we consider only three of them - grassland to evergreen needleleaved forest (ENF), grassland to deciduous broadleaved forest (DBF), and grassland to evergreen broadleaved forest (EBF) conversion. As the ESMs simulations document deforestation, we changed the signs of the observations prior to comparing them against the simulations and thus assumed that the effects of reforestation and deforestation are symmetric. Previous studies (Alkama and Cescatti, 2016; Prevedello et al., 2019) have shown that the local biogeophysical effects of afforestation and deforestation on temperature are to first order similar in magnitude but with opposite sign. While the assumption of proportionality is commonly adopted (e.g., Winckler et al., 2019a), a new study has suggested a certain degree of asymmetry in that response (Su et al., 2023). The grassland to ENF conversion is used when comparing the surface temperature response to boreal deforestation, as needleleaf trees are the predominant tree type in this region. Similarly, the grassland to DBF transition is applied for deforestation in the temperate region, and the grassland to EBF transition for deforestation in the tropics.

Figure 1. Forest cover change in the deforest-glob simulation

We also utilized a compilation of satellite-derived MODIS data products of key climate and biogeophysical variables. These include daytime land surface temperature (MYD11A2) (Wan et al., 2021), albedo together with snow cover (MCD43C3) (Schaaf and Wang, 2021) to account for the differences in background climate between ESMs and observations, and latent heat flux (MOD16A2) (Running et al., 2017). Except for the latent heat flux, for which only an older version (v006) of the MODIS data products is available, the most recent collection v061 was used (LP DAAC, 2023). Monthly data were retrieved for all years between 2003 and 2012. Similarly to other studies (e.g., Li et al., 2015), the blue-sky albedo that was used in our

analysis and considered to be representative of mean conditions, was calculated as the average of the black-sky and white-sky shortwave broadband albedo. The albedo is instantaneous, provided at local solar noon time and averaged over 16 days. For albedo and snow cover, only observations with "relative good quality" (Schaaf and Wang, 2021) or higher were considered. All MODIS-based datasets were reprojected to the WGS84 coordinate system and resampled to 0.05°.

In addition, the Landsat-based Global Forest Change product developed by Hansen et al. (2013) was used to estimate the differences in forest cover. For the period 2003-2012, the Hansen et al. (2013) dataset only reports the presence or absence of forest cover loss and gain and therefore, therefore, does not give a direct estimate of the percentage of forest cover. To retrieve the percentage of forest cover change, the pixels at the original resolution of 30 m were resampled to  $0.05^{\circ}$  using an averaging filteraveraging, thus giving us a percentage estimate of the gain/loss for each  $0.05^{\circ}$  grid cell. Because the gain in forest cover is not reported for each year but as a binary mask for the period from 2000 to 2012, a linear change was assumed to retrieve yearly values (Alkama and Cescatti, 2016). The difference between the loss and gain layers represents the forest cover change. The dataset was processed in Python using the *rasterio*, *xarray* and *rioxarray* packages (Gillies et al., 2013; Hoyer and Hamman, 2017).

# 2.3 Extracting the deforestation climate signal






In the *deforest-glob* ESM experiment, the deforestation signal in land surface temperature, albedo and latent heat flux is derived by calculating the difference between the mean of the first 30 years of the pre-industrial control simulation (*piControl*), from which *deforest-glob* is branched off, and the mean of the last 30 years of the deforestation simulation as in Boysen et al. (2020). The change in forest fraction is calculated as the difference in tree cover before and after the deforestation took place.

In the MODIS-based data products, a multi-year mean is calculated in order to diminish the effect of interannual climate variability (Baker and Spracklen, 2019). The deforestation response of the climate variables is calculated as the difference between the mean in the period 2003-2007 and the mean in the period 2008-2012. This approach is different from the one in Alkama and Cescatti (2016), which considers only pairs of single year means for the climate variables and thus does not implicitly account for interannual climate variability. The change in forest fraction is represented by the net change in forest cover in the period 2003-2012.

In the FLUXNET-based dataset, the land cover and land use change signals are calculated by adding the surface temperature responses triggered by changes in the albedo, the albedo, heat conducted by the surface medium, and the turbulent energy redistribution, which are based on monthly mean climatologies from 2001-2011 (Bright et al., 2017). The change in forest fraction is 100% and is considered only for pixels, where the respective vegetation cover types are actually present and/or could potentially occur as defined by MODIS-derived land cover maps for 2005 and Köppen-Geiger climate zone maps for the 20th century.

The differences between the various datasets in terms of temporal resolution and continuity, consideration of cloud coverage, aggregation methods and consideration of land/ocean/atmosphere interactions are summarized in Table A2.

Figure 2. Separation of local and non-local effects in ESMs. a) shows the list of predictors and predictands used as inputs in the multiple linear regression; b) the regression is trained for each pair of 5x5 grid cells moving windows; c) the methodology is applied for all three predictands; the regression coefficients  $\beta_{0,1,2,3,4}$  and the error term  $\epsilon$  are specific for each pair of moving windows;  $\beta_1$  represents the local sensitivity to tree cover change.

## 2.4 Separating local and non-local effects




The separation of local and non-local climate effects of deforestation in the ESM simulations is necessary in order to be able to compare deforestation signals in the simulations and observations, as the latter only captures local effects (Pongratz et al., 2021). In this study, we separated local and non-local effects in ESMs in a similar fashion as in the moving window approach by Lejeune et al. (2018). Alternative approaches of using information from different vegetation types at sub-grid level (Malyshev et al., 2015) or a separation through additional simulations alternating grid cells with forest cover change with unaltered vegetation cover (Winckler et al., 2017) are not applicable to the CMIP6 output.

The method by Lejeune et al. (2018) entails fitting a linear regression between the temporal changes in the climate variables and the changes in forest cover in neighboring pixels. However, instead of temporal changes, we use the climatological monthly mean values as a dependent variable from both the *piControl* and the *deforest-glob* simulations (Fig. ‡2a). This modification is necessary because with the linear rate of deforestation (as performed in the *deforest-glob* experiments) the change pattern in neighboring grid cells is too similar to determine a linear relationship between forest cover change and the variable of interest (e.g., surface temperature). With our proposed modified method, the linear regression is trained by using simultaneously the higher values of forest cover from the *piControl* simulation and the lower values of forest cover from the *deforest-glob* simulation together with the corresponding surface temperature, albedo or latent heat flux represented as a function of forest cover, thus increasing the variation in the respective climate variables and improving the robustness of the linear regression (Fig. ‡2b-c).

The exact method consists of the following steps: a moving window corresponding to  $5 \times 5$  model grid cells is applied over the variable of interest; for each window pair, a linear regression is trained using four predictor variables: tree cover, latitude,

longitude and elevation. Hereby, the linear regression is calculated only for pixels with more than 10% forest cover change similarly to Chen and Dirmeyer (2020) and only for windows where at least eight pixels are available (for consideration of signal-to-noise ratios). The slope of the tree cover variable represents the local sensitivity of a variable of interest (predictand) to deforestation. This approach is adopted for all predictand variables (land surface temperature, albedo and latent heat flux). The size of the moving window remains the same independently of the resolution of the ESM. The only exception is the IPSL model, for which the window size was adjusted to 3 grid cells in longitude and 5 grid cells in latitude to account for the higher latitudinal resolution.

The FLUXNET-based dataset provides only the local surface temperature response to land cover changes (Bright et al., 2017). For the MODIS-based datasets, the local effects are extracted using the spatial gradient method by Alkama and Cescatti (2016). Pixels with stable forest cover are identified, which are defined as having less than 2% difference in tree cover in the period 2003-2012, and the change in the respective climate variable at these locations is interpolated, so that the background climate signal can be retrieved and removed from the overall climate response. It has to should be noted that the background climate signal is not equivalent to the biogeophysical non-local effects in models as the former encompasses also natural interannual climate variability and greenhouse gas forcings. The local sensitivity to deforestation is determined by training a linear regression with zero intercept in a moving 12° by 12° window, where tree cover is the predictor variable and its slope is the sensitivity. The sensitivity represents the change in surface temperature, albedo, or latent heat flux corresponding to 1% change in tree cover.

# 2.5 Constraining local sensitivities to deforestation – emergent constraints







When comparing the sensitivities of ESMs and satellite observations, a scaling to 100% deforestation is applied to account for the different levels of tree cover change and to improve comparability with existing studies (Wang et al., 2023). Here, we assume a linear response of the climate variables to tree cover change, although a recent study suggests that afforestation and deforestation have effects with differing magnitudes (Su et al., 2023). This assumption affects mostly the satellite-based data, where the tree cover changes in both directions. Thus, depending on whether a pixel has undergone more afforestation or deforestation, the strength of the sensitivities might be underestimated or overestimated, respectively. If estimated separately for tree cover gain and loss, the difference in surface temperature sensitivity can reach 0.15°C in absolute value (Su et al., 2023).

In order to evaluate the consistency of seasonal surface temperature responses between the ESMs and the observation-based estimates, we extracted the mean values of local surface temperature responses to deforestation also at monthly timescale for broad latitudinal regions: boreal (from 50° N to 90° N), temperate (from 23° N to 50° N) and tropical (from 23° S to 23° N). We repeated this analysis step for albedo and latent heat responses to aid the interpretation of the surface temperature effects.

#### 2.6 Constraining local sensitivities to deforestation – emergent constraints

One of the key goals in this study is to provide observational-based emergent constraints (Hall and Qu, 2006) on the ESM responses to deforestation by comparing local surface temperature sensitivities and key local biogeophysical sensitivities to

changes in forest cover in observations and models. Observations cannot be directly compared against ESMs because the spatial extent, location and background climate conditions determine the biogeophysical response to deforestation and differ between the observations and simulations. However, local sensitivities are largely independent of these effects (see Introduction), allowing us to apply the emergent constraint approach. The emergent constraints concept states that a relationship between two variables can emerge across simulations with different background climate (i.e. current climate and future climate projection) in a sufficiently large ensemble of ESMs. Knowing the dependence between the two variables and having observational data for one of the variables, it is possible to constrain the second variable for which no actual observations are available (Hall and Qu, 2006). Thus, by By showing the linear relationship between local and total surface temperature change due to deforestation, we are able to constrain the overall response of ESMs to deforestation, for which no observations are available, as local surface temperature sensitivities from models and observations are comparable (Fig. 8).

can be brought into a meaningful relationship. By using the local sensitivities derived from in-situ and satellite-based data, we are potentially able to provide emergent constraints for surface temperature, albedo and latent heat flux. The sensitivities of these climate variables to deforestation are studied both temporally and spatially to account for the fact that some climate models might perform better under certain climatic conditions and/or for certain regions.

We refer to "observational constraints" as a broader term, encompassing emergent constraints. When non-local or total effects are referred to, we use specifically the term (observational) emergent constraints, as these effects can be evaluated only by considering the emergent linear relationship between the different models.

#### 3 Results






## 3.1 Spatial pattern in local responses to deforestation in observations and ESMs

Averaged over the globe, the local response to complete deforestation is mostly dominated by cooling in the northern latitudes, which overwhelms warming in the tropics. The magnitude and spatial pattern of local responses, however, vary across the ESMs with some showing weaker cooling in the northern latitudes (e.g., MPI, IPSL and BCC) compared to the rest of the models (Fig. 23). In the tropics, all models except MIROC show local warming with varying magnitude and spatial patterns (Fig. 23). The weaker cooling in some climate models (e.g., MPI, IPSL, BCC) in the boreal region is more consistent with the MODIS-based local responses to deforestation in comparison to the models showing strong non-local cooling effects (Fig. 23). A closer look into tropical regions indicates complex and partially diverging local responses to deforestation. For example, CESM and IPSL show stronger warming in the southern part of the tropical region both in the Amazon and Congo basins.

#### 3.2 Seasonal local responses to deforestation in observations and ESMs

Our results show that in the boreal region, observation-based local surface temperature responses exhibit cooling during boreal winter and warming in boreal summer (Fig. 34a). Although this seasonal pattern is confirmed by observational and model studies (Alkama and Cescatti, 2016; Strandberg and Kjellström, 2019; Winckler et al., 2019b), it is only reproduced by a

**Figure 3.** Local annual surface temperature response to complete deforestation for eleven ESMs, which have conducted the *deforest-glob* simulation (a-k). Stippling indicates non-statistically significant changes between a control and the deforestation scenario at the 5% significance level. (l) shows the local annual surface temperature response based on MODIS data. For this, the change in surface temperature is calculated as the difference between two reference periods: 2003-2007 and 2008-2012. The change in forest cover (based on Landsat data) is calculated as the cumulative change (gain minus loss) from 2003 to 2012.

subset of ESMs (MPI, IPSL and BCC), while the majority of the other models shows a cooling response throughout the year (albeit with substantially more cooling in the colder season) (Fig. AA2a). In comparison to satellite Compared to observations,

the ESMs also show a considerably stronger increase in surface albedo following deforestationcompared to observations especially in the colder season, which may explain the stronger cooling in the ESMs (Fig. 34a-b). It has to be noted that the MODIS product has a limited number of pixels in the boreal region fulfilling the quality criteria during the winter months, making the albedo monthly average values less representative compared to the other seasons. The low count of quality pixels explains the lower albedo in January compared to December and February. An additional analysis concentrating only on the grid cells, where valid MODIS pixels exist, did not reveal considerable changes in the ESMs' response (Fig. A2). Satellite observations show also Satellite observations also show a decline in the latent heat flux due to deforestationthat, which is centered in the growing season, which and may overwhelm the relatively weak albedo effect giving rise to warming during this time of the year (Fig. 34a-c). The ESMs' latent heat flux reductions due to deforestation tend to be larger compared to observations (albeit with considerable spread), but in this case the strong albedo response in the ESMs seems more important in terms of changes in the surface energy balance leading to the cooling pattern (Fig. 34a-c).

In the temperate region, the observational datasets show predominantly warming, which peaks during the boreal summer months, with the MODIS dataset exhibiting warming during the entire year, while the FLUXNET-based dataset displays weak cooling during winter (Fig. 34d). The FLUXNET-based dataset has almost constant warming during spring and summer, while the MODIS surface temperature response spikes in June. While broadleaved forests in Europe were replaced by coniferous forests throughout the last centuries and may not be the dominant forest type in Europe anymore (Naudts et al., 2016), the local surface temperature response of the two types of forest conversion is almost identical (Bright et al., 2017) and therefore the choice of a specific forest transition does not influence the observational constraint (Fig. A3). The ESMs, on average, show less cooling during June, July and August compared to the rest of the year, with only MPI, IPSL and BCC exhibiting warming during the boreal summer (Fig. AA2d). The albedo shows a pattern similar to that of the boreal forest but attenuated during the colder months (Fig. 34e). As in the boreal region, the ESMs' albedo response is considerably stronger than the MODIS one. The latent heat flux response of ESMs is also stronger compared to observations (Fig. 34f). However, not all models are able to reproduce the anticipated overall annual decrease in latent heat. In those ESMs, which do simulate the decline, and also in observations, the decrease of latent heat flux is more pronounced during spring and summer coinciding with vegetation growth.

In the tropics, most models show warming (Fig. 34g). The MODIS surface temperature signal lies considerably above the FLUXNET-based dataset and the model ensemble mean, while the latter two agree closely. The only model that shows consistent cooling over the tropics is GISS (Fig. AA2g). In the tropics, the albedo plays a secondary role in explaining the surface temperature response (Fig. 34h). The differences before and after deforestation are much smaller in magnitude and consistent throughout the year. The ESMs mean albedo is slightly higher than the observational estimate. The latent heat flux response of ESMs is stronger compared to observations and does not have seasonal fluctuations (Fig. 34i). Some models show an increase in latent heat flux as a result of deforestation. Such anomalous behavior is observed also when considering the combined effects of deforestation (Boysen et al., 2020).

**Figure 4.** Monthly local surface temperature (first row), albedo (second row) and latent heat flux (third row) responses to complete deforestation for the boreal (a, b, c), temperate (d, e, f), and tropical region (g, h, i). Only models showing statistically significant changes at the 5% significance level for all months are included (following this criterion MIROC is excluded).

#### 3.3 Comparison of local sensitivities to deforestation between observations and ESMs – emergent constraints


The largely consistent results behavior (among the ESMs, among the observations, and between the ESMs and the observations) in regards regard to local responses to deforestation in albedo and latent heat flux and the corresponding local surface temperature open up the possibility of providing observation-based emergent constraints. A key result is that across the ESMs the overall local surface temperature response to deforestation exhibits a strong linear relationship with the local surface temperature sensitivity (Fig. 8) giving rise to an emergent constraint also for albedo, as albedo sensitivity, so that models that have a strong albedo sensitivity show stronger cooling (Fig. 45a-c). This is particularly evident in the boreal and temperate regions with robust linear relationships between these metrics as shown by the percentage of variance explained in the model simulations (0.59 and 0.76 for the boreal and temperate regions, respectively). Importantly, for the boreal and temperate re-

Figure 5. Observational Emergent constraints for local annual surface temperature and albedo for the boreal (a), temperate (b) and tropical (c) regions. The dashed line with blue-orange uncertainty range shows the FLUXNET-based surface temperature sensitivity to complete deforestation (Bright et al., 2017). All sensitivities are scaled to 100% deforestation, so that the sensitivity represents the change in surface temperature/albedo corresponding to 100% change in tree cover. The error bars show the standard deviation based on annual mean values. The percentage variance explained is denoted by  $R^2$ .

gions, the local surface temperature and albedo sensitivities based on observations are considerably smaller in magnitudes compared to the ESMs and in the case of surface temperature exhibit even opposite signs (Fig. 4a,b5a-b). In the tropics, the albedo sensitivities to deforestation in the ESMs also tend to be slightly overestimated, however, still rest close to the observational constraint (with most models falling within the boundaries defined by the standard deviation of the MODIS observations) (Fig. 45c).





The discrepancy in local albedo and surface temperature sensitivities to deforestation in observations and climate models in the northern latitudes may be partially explained by different levels of observed and simulated snow cover. That is because the deforestation-induced albedo increases are thought to be largest in regions with extensive snow cover (due to the loss of effective masking of the snow albedo by darker trees (Bonan, 2008)). A complementary analysis for the boreal region during spring showed differences in snow cover extent between the ESMs and observations (Fig. 5a,b6a-b), which can in part be explained by the colder background climates in the ESMs that are more representative of preindustrial conditions (see Methods). Separating the effect of different levels of snow cover reveals that for pixels with higher snow cover in the boreal region during spring, the difference in albedo sensitivity to deforestation between climate models and observations is largest, whereas over regions with less snow cover, these differences become smaller (Fig. 56c-e). This relationship persists also when accounting for the different geographic distribution of the pixels (Fig. A3A4).

Taken together, these results do suggest that in comparison to observations the subset of CMIP6 ESMs investigated here substantially overestimate albedo increases resulting from deforestation in northern latitudes (especially in boreal regions with extensive snow cover) and as a result produce considerable local and non-local cooling responses, whereas the observations show only little changes in local surface temperature response (in part due to compensating effects of local cooling and warming responses during the colder and warmer periodsseasons, respectively).

**Figure 6.** Monthly mean boreal snow cover for models and observations (a). Distribution of boreal snow cover extent in models and observations for the spring season (March and April) (b). Emergent constraints for the boreal spring for different levels of snow cover (c, d, e). Only March and April are considered due to substantial snow cover differences between models and observation in May. The FLUXNET-based dataset (depicted with dashed line with orange uncertainty range) (Bright et al., 2017) does not contain information about snow cover, so a spring average is displayed.

Monthly mean boreal snow cover for models and observations (a). Distribution of boreal snow cover extent in models and observations for the spring season (March and April) (b). Observational constraints for the boreal spring for different levels of snow cover (c, d, e). Only March and April are considered due to substantial snow cover differences between models and observation in May. The FLUXNET-based dataset (depicted with dashed line with blue uncertainty range) (Bright et al., 2017) does not contain information about snow cover, so a spring average is displayed.

#### 3.4 Results for surface temperature and latent heat flux sensitivities to deforestation



Unlike for albedo, no clear linear relationship between local surface temperature and latent heat flux sensitivities to deforestation could be identified in the ESMs (Fig. 67). Across the ESMs, the local latent heat flux sensitivities to deforestation show a wider spread in the tropics compared to the northern regions, but many of the simulated responses are within (or not far from) the corresponding satellite-based constraints (Fig. 67).

In the tropics, the local surface temperature sensitivities to deforestation differ markedly in the satellite-based and in situ in-situ estimates. The majority of the ESMs show shows a positive local surface temperature sensitivity to deforestation well within observational constraints, but two models also show an (unexpected) negative local surface temperature sensitivity

Figure 7. Observational Emergent constraints for local annual surface temperature and latent heat flux sensitivities for the boreal (a), temperate (b) and tropical (c) regions. The dashed line with blue orange uncertainty range shows the FLUXNET-based surface temperature sensitivity to complete deforestation (Bright et al., 2017). All sensitivities are scaled to 100% deforestation, so that the sensitivity represents the change in surface temperature/latent heat flux corresponding to 100% change in tree cover. The error bars show the standard deviation based on annual mean values.

that may be explained by an overestimation of the albedo increase following deforestation (e.g., GISS, UKESM) (Fig. 45c and Fig. 67c). In other cases, models that are close to the observational constraints for surface temperature sensitivity may compensate their high albedo sensitivity with an even stronger latent heat flux sensitivity (e.g., MPI) leading to a realistic temperature response (Fig. 45c, Fig. 67c and Fig. A4A5c).





# 3.4 Local, non-local and total surface temperature effects under consideration of albedolarge-scale deforestation

While a necessary requirement for a good model is that local surface temperature sensitivities to deforestation agree with observations, an intriguing question is if deviations at local levels also translate into deviations in the total surface temperature response. A corresponding analysis comparing local and total surface temperature sensitivities to deforestation does reveal a linear relationship between these two variables giving rise to another set of emergent constraints (Fig. 8). For the boreal and temperate regions specifically, these results show that models that overestimate the local cooling effect of deforestation also tend to overestimate the total cooling effect. The interpretation of these emergent constraints, however, is not straightforward as some models that agree well with observations in regard to local surface temperature sensitivities may achieve this only because of compensatory effects of high sensitivities to albedo and latent heat (see Section 3.3).

In order to isolate the strong albedo effects on surface temperature in the northern hemisphere, a separate analysis is performed concentrating on boreal summer (June, July, August) (Fig. 79). In this analysis, we concentrate focus on models having plausible local albedo and latent heat flux sensitivities, defined here as being within two standard deviations of the model ensemble mean (as presented in Fig. 3). It has to be noted that in our study the local effect represents the surface temperature response due to incomplete deforestation as defined in the *deforest-glob* experiment. Thus, it is expected that the modeled effects are attenuated in comparison to the observational datasets. Here, the scaling to 100% deforestation is not appropriate as non-local effects cannot be directly attributed to the percentage of tree cover, although some authors suggest a linear relationship between

**Figure 8.** Emergent constraints for local and total annual surface temperature for the boreal (a), temperate (b) and tropical (c) regions. The dashed line with orange uncertainty range shows the FLUXNET-based surface temperature sensitivity to complete deforestation (Bright et al., 2017). The dashed line with blue uncertainty range shows the MODIS-based surface temperature sensitivity to complete deforestation. The local sensitivities (x-axis) are scaled to 100% deforestation, so that the sensitivity represents the change in surface temperature corresponding to 100% change in tree cover. The y-axis shows the total (local and non-local) surface temperature response to deforestation, no scaling is applied. The error bars show the standard deviation based on annual mean values.

non-local effects and the number of deforested grid cells (Winekler et al., 2019a)4), which leads to the exclusion of four out of the eleven ESMs.

In the summer, a strong local warming effect can be observed, however, this effect is not spatially homogeneous. For the boreal region and part of the northern hemisphere temperate region (up to 34°N) a local cooling is seen (Fig. 79c). A summer non-local cooling, although not as strong compared to the annual non-local effects, can be observed throughout the globe despite the albedo effect of snow being excluded (Fig. 79a). The summer non-local effects are strongest in the high latitudes (above 50°N). However, strong variation exists among ESMs, with some models showing non-local warming in the boreal region, specifically at the place of deforestation (Fig. 9A). In the tropics, there is also non-local warming at the place of deforestation, most prominent in the Amazon, though it is not strong enough to overpower the non-local cooling from neighboring grid cells (Fig. 79a,d). The compound response of summer local and non-local effects (i.e. total effect) is dominated by cooling, except for the tropics, where an overall warming can be observed (Fig. 79b). This warming is strongest in the southern part of the Amazon. The total summer surface temperature response over land averages to -0.23K globally, comprised by 0.02K local warming and -0.25K non-local cooling.

#### 4 Discussion





#### 4.1 Identification of observational constraints for surface temperature

Our study shows that climate models largely agree on the sign and the general spatial pattern of surface temperature change as a result of deforestation, however, nevertheless, the magnitude of these changes differs across models and observations.

**Figure 9.** Surface temperature differences due to deforestation during boreal summer (June, July, August) in land grid cells (a); the local and non-local effects are calculated as an average of all models having local albedo and latent heat flux sensitivities within two standard deviations of the model ensemble mean (thus, excluding CanESM, CNRM, MIROC and UKESM); here, only the actual deforestation in the *deforest-glob* experiment is considered. All lines are smoothed using a 10° moving average. (b), (c) and (d) show the total, local and non-local effects of deforestation during boreal summer. The non-local effects are calculated as the difference between the total and local effects. Only statistically significant changes at the 5% significance level are shown. All datasets are resampled to approx. 1°.

The overall pattern of warming in the tropics and cooling in the northern latitudes is in line with previous studies on the local effects of deforestation (e.g., Winckler et al., 2017). However, a more detailed look at the tropics reveals differences in the surface temperature response in the Amazon, Congo basin and Southeast Asia, which are not that pronounced in observational datasets (Fig. 23). Some of these discrepancies are thought to be triggered by differences in the spatial distribution of the

initial tree cover in the ESMs, while other stem from differences in the strength of vegetation-atmosphere feedbacks. The stronger warming that is observed in the southern part of the Amazon may be linked to the deforestation-induced strong decrease in evapotranspiration during the dry season (Zemp et al., 2017; Baker and Spracklen, 2019), which is longer and more pronounced in the southeastern part of the basin (Davidson et al., 2012). MIROC shows no significant effects on surface temperature from deforestation which may be likely as a result of the fast regrowth of forest, which is immediately merged into existing vegetation with developed canopy (Boysen et al., 2020).

Throughout the year, most models show a consistent overestimation of the cooling response in the boreal and temperate regions. Because of the emergent constraints relationship between the local and total surface temperature effects, this overestimation is valid also for the overall response to deforestation, thus showing that most models exhibit too strong cooling in comparison to observations (Fig. 8). For the tropics, however, approximately half of the ESMs show realistic total surface temperature response, as defined by the emergent constraint based on MODIS data, with fewer models being within the realistic margins defined by the FLUXNET-based dataset (Fig. 8). In the tropics, there is a better agreement between the ESMs ensemble mean and the FLUXNET-based estimate. The MODIS estimate in all regions and particularly in the tropics lies considerably above the model mean (Fig. 34). This disparity can be explained by the bias of optical remote sensing products towards cloud free days and the resulting overestimation of land surface temperature (Li et al., 2015). While this bias occurs globally, it is most notable in the tropics because of the high cloud cover fraction there. The overpass time of the Aqua satellite (on board of which is the MODIS sensor) is at 1:30 pm and thus closer to the daytime maximum surface temperature rather than the daily surface temperature average used in the model comparison. Using the daytime maximum surface temperature in the comparison with MODIS data has shown more consistent results (Chen and Dirmeyer, 2020). Accounting for cloud cover in ESMs can also make them more comparable with observations (Chen and Dirmeyer, 2020). The level at which temperature is measured (i.e. surface temperature, temperature at the lowest atmospheric layer, near-surface air temperature) also influences the strength of local effects (Winckler et al., 2019c).

#### 4.2 Identification of observational constraints for albedo





Similarly to Boisier et al. (2012), all models showed a consistent increase in albedo after deforestation with varying magnitude. In the boreal region and to a lesser extent in the temperate region, discrepancies in the albedo response in models and observations (based on MODIS) were found. These discrepancies only partially stem from the differences in snow cover conditioned by the reference climate settings. An additional analysis accounting for the different levels of snow revealed that albedo sensitivities over snow are overestimated (Fig. 56), which has been confirmed also by Lejeune et al. (2020) and Luo et al. (2023). Based on the emergent constraints (Fig. 48), the overestimated local sensitivities of albedo suggest that the overall albedo response and the corresponding cooling are also overestimated. In the boreal region, the overestimation is possibly related to the cold bias in Siberia still observed in many CMIP6 models (Portal et al., 2023). Our results revealed that some models (e.g., MPI) closer to the observational constraint for surface temperature (i.e. global mean local warming) tend to compensate their high latent heat flux sensitivity with high albedo sensitivity (Fig. A4A5). Thus, models having overestimated albedo and turbulent heat flux sensitivities can be close to the observational surface temperature constraint as compensating effects occur. Luo

et al. (2023) have also reported that models representing better surface temperature after deforestation do not necessarily have realistic albedo and turbulent heat flux estimates. The slope of the linear relationship between surface temperature and albedo sensitivity decreases with increasing snow cover, indicating a non-linear behavior of the sensitivities. Gottlieb and Mankin (2024) point out that the snow cover in spring is less affected by warming if the climatological winter temperatures are below -8°C. Thus, colder regions with more snow cover are expected to have a weaker relationship between temperature and albedo sensitivities.

## 4.3 No emergent constraints for latent heat flux







No clear linear relationship could be observed between surface temperature and latent heat flux sensitivities to deforestation (Fig. 67). While the absence of such an emergent constraints relationship may not be too surprising for the boreal and temperate regions (since latent heat changes may not yield a first-order influence on annual surface temperature responses), the apparent lack of such a relationship also for the tropical regions is surprising given the strong influence of the latent heat flux on temperature responses (Bonan, 2008). Indeed, ESMs still cannot reliably estimate the change in latent heat flux, as evidenced by the wide spread of the sensitivities and the disagreement in the sign of the change reported also in earlier studies (de Noblet-Ducoudré et al., 2012; Boisier et al., 2012; Devaraju et al., 2018; Duveiller et al., 2018a). Recent research (Winckler et al., 2019b; Devaraju et al., 2018) has explored the effects of surface roughness and the consequent changes in turbulent heat fluxes, arguing that surface roughness could be the main factor modulating the local surface temperature response even in the boreal forest during the spring season, when albedo effects are strongest. Our analysis showed that this effect reported on the basis of simulations with MPI and IPSL might be related to the strong latent heat flux sensitivity of these models (Fig. 67a-c).

The albedo and latent heat flux sensitivities shown here are in line with Devaraju et al. (2018), who report that IPSL has stronger turbulent heat flux sensitivity compared to CESM, which, on the other hand, exhibits stronger albedo sensitivity. Two of the models (EC-Earth and CMCC) show a mean increase in latent heat flux, which in the case of EC-Earth might be partly due to the replacement of trees with very productive grasses with high Leaf Area Index in wetter areas. The difficulties of ESMs in reproducing turbulent heat fluxes are well known and have also been confirmed in the newest generation of CMIP6 models (Luo et al., 2023).

#### 4.4 Non-local effects in comparison with other studies

In our analysis of the local and non-local effects during boreal summer, we show that there is non-local cooling associated with deforestation throughout the globe, which compensates for most of the local warming except for the tropics (Fig. 79). The non-local cooling could be explained with the increase of albedo and consequently the decrease of net surface radiation. While the higher albedo causes cooling both locally and non-locally, the local albedo-induced cooling is offset by decreases in latent and sensible heat fluxes (Winckler et al., 2019a). The resulting cool and dry air is moved away from the place of deforestation through advection (Winckler et al., 2019a). The non-local cooling reported here only partially agrees with Chen and Dirmeyer (2020), who also observe non-local cooling in the temperate and boreal regions, and however reveal stronger non-local warming in the tropics. This discrepancy might be explained by the fact that Chen and Dirmeyer (2020) consider

daily maximum surface temperature during cloud-free days, while the non-local effects reported here refer to mean surface temperature without being limited to cloud-free days and only account for partial deforestation. The globally averaged non-local cooling, in general, agrees with other studies (e.g., Devaraju et al., 2018; Winckler et al., 2019a). It is stronger in the mid and high latitudes, while in the tropics the local effects dominate the temperature response in line with Devaraju et al. (2018) and Winckler et al. (2019a). However, the magnitude of the non-local effects largely depends is largely dependent on the extent of deforestation (Winckler et al., 2017), thus making a comparison with other deforestation experiments difficult.

## 5 Limitations




In comparing the biogeophysical effects of deforestation between models and observations, there are a number of limitations to 460 be considered. The method used for the separation of local and non-local effects could influence the magnitude of the effects. A comparison of a spatial interpolation method commonly used in chessboard pattern deforestation experiments (Winckler et al., 2017) with the moving window approach by Lejeune et al. (2018) revealed that the latter could lead to an underestimation of local effects up to a factor of two. An additional analysis comparing temperature sensitivities to deforestation based on the linear regression method applied in this study and the chessboard pattern deforestation experiments of Winckler et al. (2019a) did not reveal evidence of systematic underestimation of local effects (Fig. A6e,f). It should be noted that in our study the 465 local effects represent responses due to incomplete deforestation as defined in the deforest-glob experiment. An additional scaling to 100% was applied in order to make these local effects comparable to observations, which only capture complete deforestation. Thus, it is expected that the modeled effects without the scaling applied (Fig. 9) are attenuated in comparison to the observational datasets. When evaluating non-local effects, scaling to 100% deforestation is not appropriate as non-local effects cannot be directly attributed to the percentage of tree cover, although some authors suggest a linear relationship between non-local effects and the number of deforested grid cells (Winckler et al., 2019a). However, scaling to 100% could lead to overestimation of local effects compared to other methods (Fig. A6g,h) possibly because the relationship between surface temperature and tree cover change is not strictly linear as albedo and evapotranspiration effects vary as a function of initial tree cover and across biomes (Bonan, 2008; Alibakhshi et al., 2020).

Another important factor are the background climate conditions. Here, we study only the biogeophysical effects of deforestation ignoring that the observations were collected in the last decades under a warmer climate compared to the pre-industrial conditions used as a reference in climate models. While we control for the differences in snow cover, we cannot account for the changes in plant physiology resulting from the adaptation to a warmer climate with higher concentration of CO<sub>2</sub> and the consequent effects on sensitivities. Traditionally, plant functional types, which capture the physiological traits of vegetation in ESMs, have been fixed (Wullschleger et al., 2014). However, the inclusion of trait variation in plant functional types as a response to environmental changes can significantly alter ESMs' outputs (Verheijen et al., 2015). Pitman et al. (2011) have also elaborated on the effects of background climate on deforestation induced changes in surface and near-surface variables, arguing that changes in rainfall and snow induced from increased CO<sub>2</sub> levels control biogeophysical effects and can even reverse their sign. It is not clear how the hydrometeorological state under increased greenhouse gases conditions affects local and non-local

biogeophysical changes separately. A simulation similar to *deforest-glob* but under fixed present-day climate conditions could improve our understanding of how background climate influences deforestation effects.

In many deforestation simulations, forested areas are converted to grasslands, which do not necessarily represent the major land use and land cover change in observations. Similarly to Devaraju et al. (2018), we assume that the different resolution of the models does not affect the separation of local and non-local effects, which does not hold true for satellite-based observations. Coarse resolution satellite measurements of surface temperature reveal a cooling in response to deforestation that is not visible in fine resolution datasets such as MODIS, which can be attributed to the fact that cloud effects are present in the coarse resolution datasets (Chen and Dirmeyer, 2020). Here, we calculate the sensitivities as latitudinal means, however, a more complete constraint analysis would include regional sensitivities as local biogeophysical effects of deforestation have distinctive regional patterns (Fig. 23).

The instantaneous Instantaneous observational measurements of albedo are often used in the modeling community (e.g., Duveiller et al., 2018a; Chen and Dirmeyer, 2020). However, these measurements do not fully correspond to the true daily mean albedo, which accounts for differences in the sun zenith angle. Daily mean albedo can be up to 8.8% higher than local noon albedo on an annual basis, and the difference can reach more than 10% under snow free conditions (Wang et al., 2015). It should also be noted that the MODIS product has a limited number of pixels in the boreal region fulfilling the quality criteria during the winter months, making the albedo monthly average values less representative compared to the other seasons. The low count of quality pixels explains the lower albedo in January compared to December and February. An additional analysis concentrating only on grid cells, where valid MODIS pixels exist, did not reveal considerable changes in the ESMs' response (Fig. A7).

Lastly, testing the emergent constraints in different experiments and multi-model ensembles is an important next step towards confirming its robustness. The overestimation of albedo sensitivities over snow and the difficulties of models in representing turbulent heat fluxes, as found in our study, have also been documented in the study of Luo et al. (2023), which is based on historical land use and land cover changes and therefore represents more realistic patterns of deforestation. Therefore, one can expect that the surface temperature emergent constraint would hold true also under more realistic conditions; however, more studies applying the emergent constraint concept in land use and land cover change scenarios are needed.

#### 510 6 Conclusions






In this study, we investigate the biogeophysical response to deforestation in eleven state-of-the-art ESMs, part of the latest CMIP6. Climate models mostly agree on the sign of the local surface temperature change after deforestation: cooling in the boreal region and warming in the tropics. In contrast, in observations, the local cooling effect is weaker and warming dominates the annual surface temperature response. For the boreal and temperate regions, the difference in the surface temperature response is stronger during the winter and spring months, mostly due to differences in the albedo. These differences can be partially attributed to the higher percentage of snow cover in climate models compared to observations. Even when accounting for the different levels of snow cover, ESMs still show stronger albedo sensitivity than observations. The robust linear

relationships of local surface temperature sensitivity with total surface temperature response, and with local albedo sensitivity point towards emergent constraints for albedo and surface temperature. Thus, the overestimation of the local albedo sensitivity and the corresponding strong local cooling are indicative of overestimation of both local and non-local effects in the ESMs in northern latitudes. The sensitivity of latent heat flux to deforestation does not show a clear relationship to surface temperature sensitivity across the different latitudes and not all ESMs reproduce the expected decrease of latent heat flux. Despite the good overlap between the ensemble mean and the observational constraint for latent heat flux, considerable variation exists between models. In some models, overestimated albedo and latent heat flux sensitivities are mutually compensated, leading to realistic surface temperature sensitivities. The local surface temperature effects of models during summer agree better with the FLUXNET-based dataset, likely because the bias towards cloud-free days is removed, and thus warming is attenuated. Strong In the summer, strong non-local effects dominate the surface temperature response in the northern hemisphere temperate and high latitudes lead to diverging local surface temperature response between climate models and observations. The non-local cooling varies regionally and persists also during summer when the effects of albedo are weaker.

The observational constraints presented here contribute further to the understanding how climate models represent deforestation and where biases exist. As models are usually evaluated based on how well they reproduce a subset of past observations, their ability to predict future climate is more uncertain (Flato et al., 2014). By using emergent constraints, modeling centers are potentially able to improve the parameterization and tuning of ESMs, so that they are better adapted to simulate future climate without being overfitted to historical data. Being aware of the limitations of ESMs can help both modelers in initiating improvements and practitioners using models to measure and maximize the efficiency of re-/afforestation efforts in mitigating anthropogenic climate change impacts.

Code and data availability. The climate model outputs are freely available from the Earth System Grid Federation (ESGF; https://aims2. llnl.gov, last access: 10 August 2023). The FLUXNET-based dataset was provided by Bright et al. (2017). The tree cover dataset is retrieved from https://storage.googleapis.com/earthenginepartners-hansen/GFC-2022-v1.10/download.html (last access: 18 August 2023). The MODIS datasets are downloaded and processed in Google Earth Engine (Gorelick et al., 2017). The code is available from the corresponding author upon reasonable request.

# Appendix A

**Table A1.** Climate variables and their respective "cmorized" names in accordance with the Climate Model Output Rewriter (CMOR) standards. The forest cover in MIROC is labeled as "forestfrac", and in GISS - as "total\_forest\_frac". All other models use the standard "treeFrac" to denote the percentage of tree cover.

| Variable name                         | CMOR name                               | Frequency |
|---------------------------------------|-----------------------------------------|-----------|
| Surface temperature                   | ts                                      | monthly   |
| Surface upwelling shortwave radiation | rsus                                    | monthly   |
| Incoming shortwave radiation          | rsds                                    | monthly   |
| Latent heat flux                      | hfls                                    | monthly   |
| Forest cover                          | treeFrac/ forestfrac/ total_forest_frac | yearly    |
| Snow cover                            | snc                                     | monthly   |

Table A2. Overview of climate variables datasets. "True" values are those that account for differences in the sun zenith angle.

|               | Timestamp                                                                                                                                                               | Temporal and spatial coverage | Temporal aggregation                                       | Land-atmosphere interactions |
|---------------|-------------------------------------------------------------------------------------------------------------------------------------------------------------------------|-------------------------------|------------------------------------------------------------|------------------------------|
| ESMs          | 3-hourly                                                                                                                                                                | Complete                      | True 3-hourly mean                                         | Coupled                      |
| FLUXNET-based | Monthly                                                                                                                                                                 | Complete                      | True 3-hourly mean                                         | Uncoupled                    |
| MODIS         | Daily at 13:30 (surface temperature); Daily mean from the acquisitions at 10:30 and 13:30 (albedo); 8-day composite from daily acquisitions at 10:30 (latent heat flux) | Clear sky conditions only     | Snapshot<br>monthly/seasonal/<br>annual clear-sky<br>means | Coupled                      |

Same as Fig. 2-3 in main text but for summer non-local effects. Non-local effects are calculated as the difference between total and local effects, where the local effects are statistically significant (see Methods). The local effects of the individual models are shown in Fig. 2-3. Same as Fig. 2-3 in main text but for summer non-local effects. Non-local effects are calculated as the difference between total and local effects, where the local effects are statistically significant (see Methods). The local effects of the individual models are shown in Fig. 2-3.

Figure A1. Observational emergent constraints for annual surface temperature for the boreal (a), temperate (b) and tropical (c) regions. The dashed line with orange uncertainty range shows the FLUXNET-based surface temperature sensitivity to complete deforestation (Bright et al., 2017). The dashed line with blue uncertainty range shows the MODIS-based surface temperature sensitivity to complete deforestation. The local sensitivities (x-axis) are scaled to 100% deforestation, so that the sensitivity represents the change in surface temperature corresponding to 100% change in tree cover. The y-axis shows the total (local and non-local) surface temperature response to deforestation. The error bars show the standard deviation based on annual mean values.

Same as Fig. 2-3 in main text but for summer non-local effects. Non-local effects are calculated as the difference between total and local effects, where the local effects are statistically significant (see Methods). The local effects of the individual models are shown in Fig. 2-3

Figure A2. Same as Fig. 3-4 in main text but with individual models highlighted.

Figure A3. Same as Fig.3 (b, e, h) 4d in main text but only including also the grassland to ENF transition for model grid cells, where MODIS pixels fulfill the quality criteria (see Methods 2.2) temperate region (Bright et al., 2017).

**Figure A4.** Same as Fig. 5–6(ac, bd, ee) in main text but only for model grid cells, where MODIS pixels fulfill the quality criteria (see Methods 2.2), thus comparing only spatially overlapping pixels in models and observations.

**Figure A5.** Albedo and latent heat flux sensitivities for the boreal (a), temperate (b) and tropical (c) regions. All sensitivities are scaled to 100% deforestation, so that the sensitivity represents the change in albedo/latent heat flux corresponding to 100% change in tree cover. The error bars show the standard deviation based on annual mean values.

**Figure A6.** Comparison between the local effects in the *deforest-glob* simulation without (a) and with scaling to 100% applied (d) and the chessboard patter simulations of Winckler et al. (2019a), where one out of four grid cells (b) or two out of four grid cells (c) are deforested. All simulations are performed with MPI-ESM.

**Figure A7.** Same as Fig. 4 (b, e, h) in main text but only for model grid cells, where MODIS pixels fulfill the quality criteria (see Methods 2.2)

Author contributions. W.B. and N.M. designed the study, and N.M. performed the data analysis. All authors contributed to the writing and revision of the text.

Competing interests. Some authors are members of the editorial board of the Earth System Dynamics journal.

Acknowledgements. The authors would like to thank the reviewers for the many valuable comments and suggestions. The EC-Earth3-Veg computations, data handling and storage were enabled by resources provided by the National Academic Infrastructure for Supercomputing in Sweden (NAISS) and the Swedish National Infrastructure for Computing (SNIC) at Tetralith, NSC, Linköping University, and SWEST-ORE/dCache, partially funded by the Swedish Research Council through grant agreements no. 2022-06725 and no. 2018-05973. P.A.M. acknowledges financial support from the strategic research area Modeling the Regional and Global Earth System (MERGE), and the Lund University Centre for Studies of Carbon Cycle and Climate Interactions (LUCCI).

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
