# Peer review of "Evaluating biogeophysical sensitivities to idealized deforestation in CMIP6 models using observational constraints"

_EGUsphere, 2025_

## Author Comment (AC2)

[Figure]

Fig. R1 Cumulative distribution of the number of pixels based on the fraction of deforestation. The magnitude of deforestation (as shown by the number of deforested pixels) in the *deforest_glob* simulation is most similar to the '*1 out of 4*' and '*2 out 4*' experiments from the study of Winckler et al. (2019), where one or two out of four grid cells, respectively, are completely deforested. Therefore, we use these two experiments for further comparison with our results.

[Figure]

Fig. R2 Comparison between the local effects in the *deforest-glob* simulation and the chessboard patter simulations of Winckler et al. (2019). All simulations are performed with MPI-ESM.

[Figure]

Fig. R3 Comparison between the local effects in the *deforest-glob* simulation with scaling to 100% applied and the chessboard patter simulations of Winckler et al. (2019). All simulations are performed with MPI-ESM.

[Figure]

Fig. R4 Forest cover change in the *deforest-glob* simulation

[Figure]

Fig. R5 Comparison between the local effects in the *deforest-glob* simulation with and without the 10% threshold being applied. All simulations are performed with MPI-ESM.

---

## Author Response (AR1)

The reviewers' comments are highlighted in blue, the responses are in red.

**Response to Review 1**

This study proposes a new methodology to separate local and non-local biogeophysical effects of deforestation in model experiments. This new methodology consists of calculating a linear regression between moving 5x5 model grids in the control simulation and the same 5x5 model grids in the deforestation simulation using tree cover, latitude, longitude, and elevation as predictor variables. The slope of the tree cover predictor variable is deemed to represent the local sensitivity of the dependent variable of interest (i.e. surface temperature, albedo, or latent heat flux). Using this new methodology on CMIP6 deforestation model experiments, this study derives emerging relationships between local and total surface temperature changes, albedo changes and latent heat flux changes. It also compares local surface temperature changes, albedo changes, and latent heat flux changes between CMIP6 model experiments and observational data from FLUXNET and MODIS to constrain model results. This study finds that the local surface temperature derived from model experiments tends to be lower than the temperature derived from observations. The albedo response is also stronger in most models compared to observations, particularly in areas with high snow cover.

This is an interesting study that provides an alternative way to calculate local effects from model experiments when it is not possible to use the checkerboard method. This provides us with a deeper understanding of the differences in local effects between models and observations. However, I believe the local effects calculated from this new methodology are contaminated by non-local effects. The extent of which is hard to determine. The magnitude of contamination would depend on the range of forest cover change among pixels included in the regression window, with a greater range likely leading to less contamination and a smaller range likely leading to more contamination. As an extreme example, if the forest cover loss is around 50% for all pixels in the window, and all pixels have a very similar surface temperature change of 1 degree between the control and the deforestation simulation, the resulting slope of the tree cover variable will be around 2. In this case, the bulk of the change in temperature would be attributed to local effects. However, this change in temperature of 1 degree encompasses both local effects and non-local effects coming from pixels outside the window. The contamination of local effects by non-local effects (which are generally a cooling) is likely one of the factors why the local temperature effects calculated by the models are significantly below observations in many cases. This should at the very least be discussed more thoroughly in the limitations and mentioned as an additional plausible cause of differences between observations and models when discussing results.

**Response to Reviewer 1 main comment about separation of local and non-local effects, and potential for underestimating the local effect of deforestation on near-surface climate:**

We'd like to point out that the moving window regression approach, which our study builds on and develops further, has been applied and evaluated by several previous studies. A very recent study (Jaeger et al., 2025, 10.1029/2024JD042698) proposed an alternative third approach to moving window regression and the chessboard approach and confirms that both are generally valid. However, there has been some evidence that the regression approach underestimates the

local effects to some extent, which, in our original submission, we acknowledge in the Limitations section (Section 5). Previous studies using a similar approach based on linear regression indeed pointed this out, when comparing this method to the factorial experiment approach (Lejeune et al., 2018) and the look-up table approach (Winckler et al., 2019 (Fig. S8)). In order to quantify the discrepancies between the different methods for separating local and non-local effects directly for our simulations, we have performed an additional analysis (see Fig. A6 in revised manuscript) comparing the temperature sensitivities to deforestation in MPI-ESM as calculated in our study based on the *deforest-glob* simulation (via linear regression) and in the study of Winckler et al. (2019), where deforestation is performed in every one or two out of four grid cells (chessboard pattern). While the deforestation patterns in these simulations are different, the magnitude of deforestation is similar and therefore it is possible to approximately compare the two methods – the one based on linear regression and the chessboard pattern method (Fig. A6). This analysis showed that the differences in temperature sensitivities between the two methods vary in geographic space and, importantly, show no systematic bias towards underestimation of the local effect (Fig. A6).

It should be kept in mind that this comparison was done only with one model (due to data availability, this comparison was possible only with the MPI model) and that the chessboard pattern method comes with its own set of uncertainties (Winckler et al. 2019). Importantly, while we cannot exclude contamination of the local effects by non-local cooling, these results suggest no systematic underestimation of the local effect as suggested by previous studies (Lejeune et al., 2018; Winckler et al. 2019). Ultimately, modellers should view our results as a first evaluation and are encouraged to perform offline simulations to further test their model's skill in replicating observationally based temperature (plus albedo- and latent heat-) sensitivities to deforestation.

Further, in this comparison (Fig. A6), we did not apply the scaling of the temperature sensitivities to 100% deforestation (see Section 2.5 in revised manuscript, lines 218-225), in order to keep the magnitude of deforestation similar to the one in the experiments of Winckler et al. (2019). Applying the scaling does lead to amplification of the differences in temperature sensitivities (resulting from the two different methods for extracting the local effects), which are more notable over the boreal region in Eurasia and the central parts of the Amazon and Congo basin (Fig. A6). However, since the same scaling approach has been applied to the observational datasets, a comparison between models and observations is still meaningful and the validity of the emergent constraints is not affected.

To respond to the specific example given by the Reviewer, where the forest cover loss for all pixels within the moving window is 50%, we would like to point out that, indeed, this pattern of deforestation has prevented us from applying the classical linear regression approach by Lejeune et al. (2018). To avoid the issue, in our proposed methodology we have taken (instead of change) the values of temperature (T) and tree cover (F), which are different in the deforest-glob and piControl simulations. And, then we train a linear regression on the pairs of T and F from both simulations.

To reflect on this Reviewer's main comment and our additional analyses, we have added the following text in the revised manuscript in the first paragraph under Limitations (Section 5):

"An additional analysis comparing temperature sensitivities to deforestation based on the linear regression method applied in this study and the chessboard pattern deforestation experiments of Winckler et al. (2019a) did not reveal evidence of systematic underestimation of local effects (Fig. A6e,f). It should be noted that in our study the local effects represent responses due to

incomplete deforestation as defined in the deforest-glob experiment. An additional scaling to 100% was applied in order to make these local effects comparable to observations, which only capture complete deforestation. Thus, it is expected that the modeled effects without the scaling applied (Fig. 9) are attenuated in comparison to the observational datasets. When evaluating non-local effects, scaling to 100% deforestation is not appropriate as non-local effects cannot be directly attributed to the percentage of tree cover, although some authors suggest a linear relationship between non-local effects and the number of deforested grid cells (Winckler et al., 2019a). However, scaling to 100% could lead to overestimation of local effects compared to other methods (Fig. A6g,h) possibly because the relationship between surface temperature and tree cover change is not strictly linear as albedo and evapotranspiration effects vary as a function of initial tree cover and across biomes (Bonan, 2008; Alibakhshi et al., 2020)."

Additionally, although the manuscript is generally well-written, it could benefit from some reorganization to improve the flow, some clarification of concepts in certain areas, and consistency in the terms used to avoid confusion, as indicated in the specific comments. Moreover, the section on non-local effects is not directly relevant to the research objectives (comparing the local effects across a range of ESMs and against observations) and could be moved to the supplementary material. The effectiveness of the figures could also be improved, mainly by making the differences between observations and model experiments stand out more, as per the specific comments.

Regarding the section on non-local effects, we have elaborated in the specific comments on the importance of considering the non-local effects. In this section, we show a way forward how to include in the comparison models that overestimate the albedo-induced cooling.

**Specific comments:**

14: Typo, northern latitudes

Lines 14: fixed in the revised manuscript

15: Unclear what emergent constraint means without having read the paper, suggest to remove "via an emergent constraint" in the abstract, or elaborate further on the concept of emerging constraints

15: suggested improvement: "[...] via an emergent constraint (i.e. resulting from the linear relationships within the model ensemble)"

55-57: Large-scale deforestation also triggers changes in ocean circulation which would impact non-local effects (Portmann et al., 2022)

55-57: suggested improvement: "But large-scale deforestation in these experiments also triggers strong changes in advection of heat and moisture, as well as in atmospheric and ocean circulation, which influence regions that have not undergone deforestation (Winckler et al., 2017; Portmann et al., 2022)."

75: Typo, cloud cover instead of forest cover

75: to be removed to avoid confusion: "[...] temperature changes resulting from changes in forest cover"

91-92: Define what observationally based emergent constraints are. Needs more background. See my comments for line 215-230. It is hard to understand what it means and if it is any different than observational constraints or simply emergent constraints?

91-92: to be removed to avoid confusion: "The objective is to provide observationally based emergent constraints..."; clarification of "emergent constraint" added (see 15)

164-166: Add how the change in forest fraction is represented when using the FLUXNET data to make it consistent with the ESM experiment and the MODIS-based data explanations

164-166: suggested improvement: "The change in forest fraction is 100% and is considered only for pixels, where the respective vegetation cover types are actually present and/or could potentially occur as defined by MODIS-derived land cover maps for 2005 and Köppen-Geiger climate zone maps for the 20th century."

185: Add comma after the word pair

185: fixed in the revised manuscript

186: You need pixels with greater than 10% forest cover change to calculate the local effects, but pixels with zero or minimal forest cover change can help strip out the non-local effects (as changes in those would be attributed mainly to non-local effects). You could therefore calculate the linear regression *for all pixels*, but *only* in windows where there is a sufficient number of pixels with greater than 10% forest cover change. It would be interesting to see how this changes the magnitude of the local effects.

186: It is not expected that including pixels with less than 10% forest cover change will alter the results substantially, as in most models the forest cover change is much larger than 10% (see revised manuscript, Fig. 1). We have provided an additional analysis comparing the surface temperature sensitivity with and without the 10% threshold for the MPI model (see Appendix at the end of this document, Fig. 1). The MPI model was selected based on the availability of more pixels with less than 10% forest cover change in comparison to other models. The results with using the threshold are more robust, especially for the boreal forest, as evidenced by the number of statistically significant pixels.

192: Consider adding that you then calculate the mean value of the local sensitivity over the period and region of interest for completeness.

192: The information is already (partially) provided in 211-214. Suggested improvement: "In order to evaluate the consistency of seasonal surface temperature responses between the ESMs and the observation-based estimates, we extracted the mean values of local surface temperature responses to deforestation also at monthly timescale for broad latitudinal regions: boreal (from 50° N to 90° N), temperate (from 23° N to 50° N) and tropical (from 23° S to 23° N)."

203: Consider starting the new subsection here "2.5 Comparing local sensitivities between observations and ESMs – emergent constraints"

203: Subsection 2.5. updated accordingly

215-230: Most of this section should go in the Introduction as it provides background info on the emergent constraint approach and how it can be used, and discusses goals of the study.

215-230: Explanation of the emergent constraint concept is moved to Introduction.

223-230: Hard to understand, would benefit from a clearer description of what you are trying to achieve: are you saying that if we know the local effects from observations, and the emerging relationship between local effects and total effects from model experiments, we could use that relationship to constrain the total effects based on the observed local effects? Regarding albedo, if we know the albedo change from observation, and the emerging relationship between albedo and total temperature from model experiments, we could use that relationship to constrain the total temperature change based on the observed albedo change?

223-230: suggested additions/changes: "By knowing the local effects from observations, and the emerging relationship between local and total effects from model experiments, we could use that relationship to constrain the total effects based on the observed local effects. Thus, by using the local sensitivities derived from in-situ and satellite-based data, we are potentially able to provide emergent constraints for surface temperature, albedo and latent heat flux. In addition, we use the relationship between local surface temperature and albedo, and between local surface temperature and latent heat flux, to explore further the emergent constraints."

227: I don't think we can say that local temperature sensitivities are comparable between models and observations in boreal and temperate regions based on your data (although they appear to be comparable in the tropics). Most models show local temperature effects significantly lower than observations, which, as discussed earlier, could be due to the incorporation of the non-local cooling effects in the local effects calculated from models.

227: The possibility to compare local effects in models with observations has been confirmed by a number of studies (e.g., Bright et al., 2019; Winckler at al., 2019a; Chen&Dirmeyer, 2020). The issue with the possible overestimation of local effects is discussed in the first paragraph. To avoid a misunderstanding about the term "comparable", we have changed the sentence from "[...] as local surface temperature sensitivities from models and observations are comparable" to "[...] as local surface temperature sensitivities from models and observations can be brought into a meaningful relationship".

229: Local surface temperature or total surface temperature?

229: Here we refer to the total effects.

249-253: Sentence starting with "It has to be noted..." to sentence ending by Fig A4: could go in the limitations section of the Discussion.

249-253: Moved to Limitations in the revised manuscript.

278: Clarify local or total surface temperature.

278: Here we refer to local surface temperature.

280: The emerging linear relationship between local temperature and total temperature is a key result, I think the figure showing this result (currently A1) should be incorporated in the main text and not the supplementary information.

280: Fig. A1 is moved to the main text in the revised manuscript (Fig. 8).

288: Add "defined by the standard deviation of the MODIS observations" for clarity.

288: Changed to "defined by the standard deviation of the MODIS observations"

304: Section 3.4 should really be part of section 3.3 and not a separate section.

304: Section 3.4 is merged with Section 3.3

316: This section on the non-local effects doesn't add much information that is directly related to the objectives of the study and can deter from the core message. I would move Figure 7 and corresponding information to the supplementary material.

316: While in this section we report also on non-local effects, which are not the primary focus of the study, we also evaluate the local effects during summer. This evaluation is particularly important because by considering only summer months we are able to exclude the effects of the Siberian cold bias still observed in many CMIP6 models (Portal et al., 2023). It is also relevant in light of the discussion on the contamination of local effects by non-local effects. In the summer, the non-local cooling is weaker (due to the absence of the snow albedo effect), thus giving possibly clearer view on the local effects.

337: Observational constraints, observational emergent constraints, or emergent constraints? Is there a difference between the three?

337: We refer to observational constraints as a broader term, encompassing emergent constraints. When non-local or total effects are referred to, we use specifically the term (observational) emergent constraints, as these effects can be evaluated only by considering the emergent linear relationship between the different models. "Emergent constraints" and "observational emergent constraints" are synonymous. Clarification is added under Section 2.5.

337: This section would also benefit from a discussion on how we could use the emergent linear relationship between local temperature and total temperature changes to constrain total temperature changes from ESMs based on observed local temperature changes and the limitations/pitfalls of doing so.

337: suggested addition at line 360: "Because of the emergent constraint relationship between the local and total surface temperature effects, this overestimation is valid also for the overall response to deforestation, thus showing that most models exhibit too strong cooling in comparison to observations (Fig. 8). For the tropics, however, approximately half of the ESMs show realistic total surface temperature response, as defined by the emergent constraint based on MODIS data, with fewer models being within the realistic margins defined by the FLUXNET-based dataset (Fig. 8)."

360: Observational constraints, observational emergent constraints, or emergent constraints?

360: see above

378: Observational constraints, observational emergent constraints, or emergent constraints?

378: see above

396: This section could be removed, since it does not add much new information compared to other studies, and if the results regarding the non-local effects are moved to supplementary material as per prior comment.

396: see comment to 316

412-496: This is a major limitation as discussed in my general comments. Indication that non-local effects contaminate local effects should be mentioned along with its repercussions, including overestimation of the local cooling effect.

412-496: see general comments

Fig 3: Replace Bright et al. (2017) by FLUXNET in the legend to make it clearer where this data come from when looking only at the figure. It is already indicated in the text that the FLUXNET data comes from Bright et al. (2017). Y axis: add "local" temperature change to avoid confusion.

Fig. 3: updated in revised manuscript

Fig 4: Use a different color for the MODIS data point (observation) to make it stand out more compared to model data. Y axis: add "local" temperature change to avoid confusion. Replace observational constraints with "observational emergent constraints" in the caption to make it consistent across all figures.

Fig 4: updated in revised manuscript

Fig 5: Same comments as Fig 4. Consider using the same color for the MODIS data point in plots c-d-e and for the MODIS lines in plots a-b, and same color for the CMIP6 data points in plots c-d-e and for the CMIP6 lines in plots a-b for consistency.

Fig 5: updated in revised manuscript

Fig 6: Same comments as Fig 4.

Fig 6: updated in revised manuscript

Fig. A5 and A6: Same comments as Fig 4, as applicable.

Fig. A5 and A6: updated in revised manuscript

**References**

Portmann, R., Beyerle, U., Davin, E., Fischer, E. M., De Hertog, S., & Schemm, S. (2022). Global forestation and deforestation affect remote climate via adjusted atmosphere and ocean circulation. *Nature Communications*, *13*(1), 5569. https://doi.org/10.1038/s41467-022-33279-9

**References**

Alibakhshi, S., Naimi, B., Hovi, A., Crowther, T. W., & Rautiainen, M. (2020). Quantitative analysis of the links between forest structure and land surface albedo on a global scale. *Remote Sensing of Environment*, 246, 111854.

Bonan, G. B. (2008). Forests and climate change: forcings, feedbacks, and the climate benefits of forests. *science*, *320*(5882), 1444-1449.

Bright, R. M., Davin, E., O'Halloran, T., Pongratz, J., Zhao, K., & Cescatti, A. (2017). Local temperature response to land cover and management change driven by non-radiative processes. *Nature Climate Change*, *7*(4), 296-302.

Jäger, F., Schwaab, J., Bukenberger, M., De Hertog, S. J., & Seneviratne, S. I. (2025). Spectral decomposition and signal separation of climate responses to land cover changes. *Journal of Geophysical Research: Atmospheres*, 130(5), e2024JD042698.

Lejeune, Q., Davin, E. L., Gudmundsson, L., Winckler, J., & Seneviratne, S. I. (2018). Historical deforestation locally increased the intensity of hot days in northern mid-latitudes. *Nature Climate Change*, 8(5), 386-390.

Winckler, J., Lejeune, Q., Reick, C. H., & Pongratz, J. (2019). Nonlocal effects dominate the global mean surface temperature response to the biogeophysical effects of deforestation. *Geophysical Research Letters*, 46(2), 745-755.

Portal, A., d'Andrea, F., Davini, P., Hamouda, M. E., & Pasquero, C. (2023). Atmospheric response to cold wintertime Tibetan Plateau conditions over eastern Asia in climate models. *Weather and Climate Dynamics*, *4*(3), 809-822.

Portmann, R., Beyerle, U., Davin, E., Fischer, E. M., De Hertog, S., & Schemm, S. (2022). Global forestation and deforestation affect remote climate via adjusted atmosphere and ocean circulation. *Nature Communications*, *13*(1), 5569. https://doi.org/10.1038/s41467-022-33279-9

**Response to Review 2**

The work of Mileva et al. explores and quantifies relationships between changes to key surface energy balance variables and forest cover change in several independent observational datasets as well as in ten distinct ESMs. From this, the authors identify robust statistical relationships between surface albedo change and local temperature change, and between local temperature change and total temperature change "emerging" from the ESMs. These findings are important when viewed in the context of similar relationships seen in the observational records since this provides insights critical to the goal of model development/improvement. The paper is well-written with logical organization, and the work has been carried out thoroughly and carefully. The methods are sufficiently described and documented, and the study's limitations are made clear.

I only have a few comments (or rather, suggestions) for improving the methodological clarity and depth of the discussion. The first is related to the methods. I think it should be made clearer upfront about the limitations of the comparisons being made between the ESM results and the observation-based datasets. "Apples-to-apples" comparisons are never really being made. Perhaps adding a table resembling the following is most efficient at clearly communicating important differences among the datasets being compared in the study.

|                   | Timestamp                              | Conditional | Aggregation                                            | Physics of ΔLST |
|-------------------|----------------------------------------|-------------|--------------------------------------------------------|-----------------|
| ESMs              | 3-hourly                               | None        | True monthly/seasonal/annual means                     | Coupled         |
| FLUXNET-
based | Monthly                                | None        | True monthly/seasonal/annual means                     | Uncoupled       |
| MODIS             | Local noon
(albedo); 13:30
(LST) | Clear skies | Snapshot
monthly/seasonal/annual
clear-sky means | Coupled         |

We thank the reviewer for his/her comments and appreciate he/she finds our study interesting.

**Response to Reviewer 2 first main comment about improving the methodological clarity:**

In order to highlight the differences between the various datasets, Table A1 was updated (see revised manuscript, Table A1), and the following table was added in the revised manuscript:

Table A2. Overview of climate variables datasets. "True" values are those that account for differences in the sun zenith angle.

|               | Timestamp                                                                                                                                                                                    | Temporal and spatial coverage | Temporal aggregation                                       | Land-atmosphere interactions |
|---------------|----------------------------------------------------------------------------------------------------------------------------------------------------------------------------------------------|-------------------------------|------------------------------------------------------------|------------------------------|
| ESMs          | 3-hourly                                                                                                                                                                                     | Complete                      | True 3-hourly mean                                         | Coupled                      |
| FLUXNET-based | Monthly                                                                                                                                                                                      | Complete                      | True 3-hourly mean                                         | Uncoupled                    |
| MODIS         | Daily at 13:30 (surface
temperature); Daily
mean from the
acquisitions at 10:30 and
13:30 (albedo); 8-day
composite from daily
acquisitions at 10:30
(latent heat flux) | Clear sky
conditions only  | Snapshot
monthly/seasonal/
annual clear-sky
means | Coupled                      |

Additionally, the following text was added to Section 2.1: "All models couple land, atmosphere and ocean in terms of momentum, matter and energy (Lawrance et al., 2016)."

Addition to section 2.3: "The differences between the various datasets in terms of temporal resolution and continuity, consideration of cloud coverage, aggregation methods and consideration of land/ocean/atmosphere interactions are summarized in Table A2."

Regarding the discussion, I wonder if the paper would be strengthened by expanding on the application benefits of the emergent constraint found in this study as it relates to model development. For example, how, specifically, would it contribute to a "deeper understanding of how local and non-local biogeophysical effects are represented in ESMs" as is stated in the Abstract? The authors demonstrate important offsetting biases in a few of the models surrounding the mechanisms governing the local surface temperature response, and thus there is a risk that such an emergent constraint might not lead to any meaningful model improvement.

**Response to Reviewer 2 second main comment about improving the discussion on emergent constraints:**

Many processes relevant for the climate cannot be explicitly represented by mathematical expressions, which require the parameterization of climate models. The development of parameterizations and the final parameter adjustment (or 'model tuning') rely heavily on observations and are two areas where the availability of emergent constraints could lead to new developments and improvements. Suggested addition to the revised manuscript (at lines 496-499): "As models are usually evaluated based on how well they reproduce a subset of past observations, their ability to predict future climate is more uncertain (Flato et al., 2013). By using emergent constraints, modelling centres are potentially able to improve the parameterization and tuning of ESMs, so that they are better adapted to simulate future climate without being overfitted to historical data."

Additionally, I feel it could also be beneficial to add some discussion surrounding the robustness of the finding of the linear relationship between the local and the total surface temperature change due to deforestation, and whether the authors think this relationship would hold in the case of more realistic (or real world) patterns and scales of deforestation.

**Response to Reviewer 2 third main comment about separation of local and non-local effects:**

See detailed responses to Reviewer 1 main comment.

Further, the magnitude of deforestation in the deforest-glob simulation amounts to 20 million km2 and is therefore similar to historical deforestation from 800 to 2015 (~22 million km²) (Boysen et al., 2020). The overestimation of albedo sensitivities over snow and the difficulties of models to represent turbulent heat fluxes, as found in our study, have been documented also in the study of Luo et al. (2023), which is based on historical land use and land cover changes and therefore represents more realistic patterns of deforestation. Taking into account also the similar local surface temperature response that can be observed in our study and the study of Luo et al. (2023) and the similar magnitude of deforestation, one can postulate that the surface temperature emergent constraint would hold true also under more realistic conditions. Testing the emergent constraint in different experiments and multi-model ensembles is an important next step towards confirming its robustness. To reflect on this comment, the following text was added to the revised manuscript: "Lastly, testing the emergent constraints in different experiments and multi-model ensembles is an important next step towards confirming its robustness. The overestimation of albedo sensitivities over snow and the difficulties of models to represent turbulent heat fluxes, as found in our study, have been documented also in the study of Luo et al. (2023), which is based on historical land use and land cover changes and thus represents more realistic patterns of deforestation. Therefore, one can expect that the surface temperature emergent constraint would hold true also under more realistic conditions, however, more studies applying the emergent constraint concept in land use and land cover change scenarios are needed."

**Detailed comments**

P5, L142-143: Since this sentence is lumped into the paragraph describing the remote sensing datasets, clarify that "all" here includes the FLUXNET-based dataset presented in the previous paragraph.

P5, L142-143: This sentence refers only to the MODIS products. Suggested change: "All MODIS-based datasets were reprojected to the WGS84 coordinate system and resampled to 0.05°."

P6, L147: "using an averaging filter". What is this (or how is it different than averaging)?

P6, L147: To avoid confusion, "filter" is removed: "[...] the pixels at the original resolution of 30 m were resampled to 0.05° using averaging"

Section 3.1 and Figure 2: The authors acknowledge the clear-sky bias limitations of observations based on satellite remote sensing (P3, L73), so I don't understand why the models' all-sky responses are being compared to the MODIS-based response? Wouldn't it make more sense to compare to the FLUXNET-based dataset here which the authors acknowledge (P3, L75) overcomes this important limitation?

Section 3.1 and Figure 2: The FLUXNET-based dataset only provides changes in surface temperature but not in albedo and latent heat flux, which are available from MODIS observations

and are essential for explaining the temperature response. Therefore, it was important to include the MODIS datasets from the beginning of the analysis. Here, we also wanted to present an updated version of local surface temperature effects as shown in the study of Alkama&Cescatti (2016), which is also based on MODIS data, by using the newest generation of these products (v061), which have undergone various calibration improvements. Detailed maps of the FLUXNET-based temperature responses are available in Bright et al. (2017).

Figure 2: I don't see any stippling on any of the figure panels.

Figure 2: All figures are to be provided at 330 dpi resolution, so that the stippling is visible (minimum required is 300 dpi).

P17, L388: I'm confused by this sentence, as these two models seem to be among those with the weakest latent heat flux sensitivity?

P17, L388: Here we refer mainly to Fig. 7a in the revised manuscript, where MPI and IPSL are among the models with the highest latent heat flux sensitivity (together with UKESM and GISS) and thus further away from the observational constraint for latent heat flux, depicted on the x-axis.

Figure A3: Consider adding the observations here as in Fig. 3 so it becomes easier for the reader to more easily benchmark the individual models.

Figure A3: updated in revised manuscript (now Fig. A2)

**References**

Alibakhshi, S., Naimi, B., Hovi, A., Crowther, T. W., & Rautiainen, M. (2020). Quantitative analysis of the links between forest structure and land surface albedo on a global scale. *Remote Sensing of Environment*, 246, 111854.

Alkama, R., & Cescatti, A. (2016). Biophysical climate impacts of recent changes in global forest cover. *Science*, 351(6273), 600-604.

Bonan, G. B. (2008). Forests and climate change: forcings, feedbacks, and the climate benefits of forests. *science*, *320*(5882), 1444-1449.

Boysen, L., Brovkin, V., Pongratz, J., Lawrence, D., Lawrence, P., Vuichard, N., ... & Lo, M. H. (2020). Global climate response to idealized deforestation in CMIP6 models. *Biogeosciences Discussions*, 2020, 1-35.

Bright, R. M., Davin, E., O'Halloran, T., Pongratz, J., Zhao, K., & Cescatti, A. (2017). Local temperature response to land cover and management change driven by non-radiative processes. *Nature Climate Change*, 7(4), 296-302.

Flato, G., Marotzke, J., Abiodun, B., Braconnot, P., Chou, S. C., Collins, W., ... & Rummukainen, M. (2014). Evaluation of climate models. In *Climate change 2013: the physical science basis.* Contribution of Working Group I to the Fifth Assessment Report of the Intergovernmental Panel on Climate Change (pp. 741-866). Cambridge University Press.

Luo, X., Ge, J., Guo, W., Cao, Y., Liu, Y., Chen, C., & Yang, L. (2023). An evaluation of CMIP6 models in representing the biophysical effects of deforestation with satellite-based observations. *Journal of Geophysical Research: Atmospheres*, *128*(12), e2022JD038198.

Winckler, J., Lejeune, Q., Reick, C. H., & Pongratz, J. (2019). Nonlocal effects dominate the global mean surface temperature response to the biogeophysical effects of deforestation. *Geophysical Research Letters*, *46*(2), 745-755.

**Response to Review 3**

The study uses simulations from eleven Earth System Models (ESMs) participating in CMIP6 to investigate the sensitivities of local and non-local biogeophysical effects to idealised global deforestation. Specifically, the authors analyse how surface temperature, albedo, and latent heat flux respond locally and non-locally to forest loss. The study introduces a modified regression-based method tailored to the linear deforestation experiment setup to disentangle these effects. Furthermore, the emergent constraint framework is applied to compare the simulated biogeophysical effects with observational evidence from satellite and flux tower data. The study finds that while ESMs agree on the general spatial pattern of deforestation-induced temperature changes (cooling at high latitudes and warming in the tropics), they tend to overestimate the cooling effect.

The manuscript is well written and uses adapted methods in a novel way to provide new insights into ESM-simulated biogeophysical responses to deforestation.

**General comment:**

A more thorough explanation or demonstration of the approach used to separate local and non-local effects would strengthen the manuscript. It remains unclear to me how the approach achieves the full separation of local effects (within the window) and non-local effects of deforestation outside the window. How can it be ensured that deforestation outside the moving window does not influence the climate variables within it, thereby affecting the linear fit that is interpreted as purely "local"? If non-local effects are not fully excluded, couldn't it affect the validity of the emergent constraint formed between local and total temperature changes? Since the separation is a key aspect of the study, a more detailed discussion of the validity of this method would enhance the robustness of the results and better establish the developed approach.

We thank the reviewer for his/her comments and appreciate he/she finds our study interesting.

Response to Reviewer 3 main comment about separation of local and non-local effects, and potential for underestimating the local effect of deforestation on near-surface climate:

See detailed responses to Reviewer 1 main comment.

In addition, it should be noted that the key idea of the separation method(s) is that the nonlocal effects are part of the background climate that is similar in close-by grid cells. Deforestation outside the moving window undoubtedly affects the climate variable within, but not in a systematic way (e.g. as a function of tree cover change within the moving window) and therefore related effects can be excluded.

**Specific comments:**

Lines 130 – 133: The assumption that a grassland to DBF transition is adequate to represent all deforestation in the temperate regions seems questionable. E.g. in Europe, coniferous forests are at least if not more prevalent (Naudts et al. 2016, https://doi.org/10.1126/science.aad7270).

**Response to specific comments of Reviewer 3:**

Lines 130 – 133: The local surface temperature response of the two transitions – grassland to evergreen needleleaved forest (ENF) and grassland to deciduous broadleaved forest (DBF), is almost identical (Bright et al., 2017, Fig. 3 d,e). Choosing the ENF instead of the DBF transition will, therefore, not influence the validity of our results for the temperate region. To reflect on this comment, the following text will be added to the revised manuscript: "While broadleaved forests in Europe were replaced by coniferous forests throughout the last centuries and may not be the dominant forest type in Europe anymore (Naudts et al., 2016), the local surface temperature response of the two types of forest conversion is almost identical (Bright et al., 2017) and therefore the choice of a specific forest transition does not influence the observational constraint." We have also provided in the Appendix of the revised manuscript an additional plot (similar to Fig. 3b in the original manuscript), showing both transitions (Fig. A4):

Figure 1: How is the green area defined?

Figure 1: The green area is defined by the slope of the multiple linear regression and is represented as a plane in the multidimensional space, defined by the predictor variables (tree cover, latitude, longitude and elevation).

Lines 277 – 279. I urge the authors to be more explicit about what they precisely mean by consistency. E.g. consistency in shape/behaviour or magnitude? I feel the broadly stated consistency over all hardly holds if looked at more individually. E.g. there are considerable differences in magnitude between models and observations for temperature and albedo. Further, there are quite considerable differences in shape in latent heat fluxes between ESMs.

Lines 277 – 279: The emergent constraint relies mostly on the consistent behaviour of the ESMs (e.g., decrease of temperature with increasing albedo), but also on the presence of variation/errors in the models (Williamson et al., 2021). For example, if all ESMs exhibit the same error or the spread between models is removed (e.g., due to model improvements), the emergent constraint could disappear (Hall et al., 2019). Suggested improvement: "The largely consistent behaviour..."

Lines 318 – 320. Does an approach working with two standard deviations make sense for a study of 11 models? Assuming a normal distribution, wouldn't that only lead to about half a model outside 2sd on average? It would be helpful if the authors could state explicitly how many models were excluded.

Lines 318 – 320: The main idea here was to provide a plausible ensemble mean by excluding outlier models. The excluded models are MIROC, CanESM, CNRM and UKESM (Fig. 9 in revised manuscript). Suggested improvement: "In this analysis, we concentrate on models having plausible local albedo and latent heat flux sensitivities, defined here as being within two standard deviations of the model ensemble mean, which leads to the exclusion of four out of the eleven ESMs..."

**References**

Alibakhshi, S., Naimi, B., Hovi, A., Crowther, T. W., & Rautiainen, M. (2020). Quantitative analysis of the links between forest structure and land surface albedo on a global scale. *Remote Sensing of Environment*, 246, 111854.

Bonan, G. B. (2008). Forests and climate change: forcings, feedbacks, and the climate benefits of forests. *science*, *320*(5882), 1444-1449.

Bright, R. M., Davin, E., O'Halloran, T., Pongratz, J., Zhao, K., & Cescatti, A. (2017). Local temperature response to land cover and management change driven by non-radiative processes. *Nature Climate Change*, 7(4), 296-302.

Hall, A., Cox, P., Huntingford, C., & Klein, S. (2019). Progressing emergent constraints on future climate change. *Nature Climate Change*, 9(4), 269-278.

Ma, H., Crowther, T. W., Mo, L., Maynard, D. S., Renner, S. S., Van den Hoogen, J., ... & Parada-Gutierrez, A. (2023). The global biogeography of tree leaf form and habit. *Nature Plants*, 9(11), 1795-1809.

Naudts, K., Chen, Y., McGrath, M. J., Ryder, J., Valade, A., Otto, J., & Luyssaert, S. (2016). Europe's forest management did not mitigate climate warming. *Science*, *351*(6273), 597-600.

Williamson, M. S., Thackeray, C. W., Cox, P. M., Hall, A., Huntingford, C., & Nijsse, F. J. (2021). Emergent constraints on climate sensitivities. *Reviews of Modern Physics*, 93(2), 025004.

Winckler, J., Lejeune, Q., Reick, C. H., & Pongratz, J. (2019). Nonlocal effects dominate the global mean surface temperature response to the biogeophysical effects of deforestation. *Geophysical Research Letters*, *46*(2), 745-755.

**Appendix**

Fig. 1 Comparison between the local effects in the *deforest-glob* simulation with and without the 10% threshold being applied. All simulations are performed with MPI-ESM.

---

## Referee Report (RR1)

The authors provided thoughtful responses to the reviewers comments to improve the clarity of the manuscript and address concerns about the separation of local and non-local effects. I particularly appreciate the comparison of the local effects between the moving window regression and chessboard methods using the MPI-ESM (Fig A6). However, I still have concerns over the overestimation of the local cooling effects in the manuscript and the implications of such overestimation for the findings.

More specifically, in the limitations section of the Discussion, the authors mention that they did not find evidence of systematic underestimation of local effects (without scaling), but that scaling to 100% could lead to overestimation of local effects compared to other methods. When performing the chessboard calculation of local effects, I'm assuming a 100% deforestation was performed on each deforested grid cell when deforesting 1 out of 4 or 2 out of 4 grid cells (Fig. A6). Given that the deforest\_glob experiment only does partial deforestation in most grid cells, the deforest\_glob scaled to 100% appears to be the best comparison with the chessboard experiment. I am questioning whether the comparison without scaling is relevant (in this case, for a given deforested grid cell, you would be comparing local effects resulting from partial deforestation from the deforest glob experiment with local effects resulting from complete deforestation in the chessboard experiment)? If local effects were comparable across methods, shouldn't we see stronger local effects in the chessboard experiment in this case? When comparing the scaled deforest\_glob experiment and the chessboard experiment, there is evidence of overestimation of the local effects as pointed out by the authors. The authors acknowledge factors that could contribute to the overestimation of local effects due to scaling, but isn't it possible that the choice of methodology itself (moving window regression vs chessboard) could lead to an overestimation of local effects? I don't think we can say that it is only scaling that leads to the overestimation of local effects.

Furthermore, the implications of such overestimation of local effects (regardless of where it comes from) for the findings are not discussed. How would that influence the emergent constraints discussed previously? Could this result in any changes in the relationships between the variables considered (local temperature, total temperature, albedo, latent heat flux)? For example, if the local effects are overestimated, the consistent overestimation of the local cooling response of the models in the boreal and temperate regions compared to observations may not be as pronounced as suggested in this study. Would that also explain part of the local albedo sensitivity which can't be explained by the higher percentage of snow cover in ESMs? Similarly, the authors also mention that the total cooling resulting from deforestation is overestimated in ESMs due to overestimation of the local cooling by ESMs (compared to observations), but that claim may also change if local effects are overestimated in this study. For example, if local effects are overestimated, it is

possible that non-local effects are underestimated (by a similar amount since a proportion of the cooling is just moved from local to non-local) which would then impact the relationship between local and total effects. Based on the evidence provided, particularly the stronger snow coverage in ESMs compared to observations, local (and total) cooling in northern latitudes do appear to be overestimated in ESMs compared to observations, but it may not be as pronounced as the results in this study are showing. Adding a discussion of the implications of an overestimation of local effects as part of the limitations would be important.

I also have a few additional specific comments, mainly to enhance clarity.

Line 15: Consider specifying the linear relationships between what variables in that sentence (i.e. resulting from the linear relationships between X and Y within the model ensemble).

Lines 89-106: Those are two important paragraphs setting the stage for the study. The stated goals of the study do not mention looking at the emerging relationship between local and total effect despite this being an important part of the manuscript. The stated goals focus mainly on the relationship between local temperature and biogeophysical properties within observations and models despite mentioning that knowing the emerging relationship between local and total effects within models would be useful. There seems to be a disconnect there.

Line 178: extra comma

Line 230-232: That sentence is not clear, maybe it should be rephrased. It also does not flow well with the end of the last paragraph; it is missing a transition. It is not clear how the local surface temperature sensitivities are related to the relationship between local and total temperature in this sentence.

Line 330: The link between this paragraph and the previous one where we talk about the relationship between local and total effect is not clear. Consider providing a better transition and elaborating on how Fig 9 ties to Fig 8.

Line 359: Would that still be the case if local effects are overestimated in this study, as per the main comment above?

Line 487: overestimation of local and total effects, not local and non-local effects. The results did not discuss overestimation of non-local effects.

Fig 5: FLUXNET-based surface temperature sensitivity should be green to match the green line of FLUXNET in Fig 4. Keep orange for MODIS.

Fig 6: Same comment as Fig 5. Also, why is Modis showing a negative local temperature change for all levels of snow cover, whereas it is positive in Fig 5 for the boreal region?

Fig 7: Same comment as Fig 5.

Fig 8: To be consistent with the colors chosen in other figures, MODIS sensitivity uncertainty range should be orange and FLUXNET sensitivity uncertainty range should be green.

Fig A4: Same comment as Fig 5.

Fig A6: pattern, typo in the caption. See also main comment regarding comparison between deforest\_glob (not scaled) and chessboard experiments.

---

## Author Response (AR2)

The reviewers' comments are highlighted in blue, the responses are in red.

**Response to Report #1 (Referee #2)**

New table "XX" added following Table A1: For the "Aggregation" column I believe the text for the FLUXNET row should read "true monthly mean".

Table A2 has been adjusted accordingly.

**Response to Report #2 (Referee #1)**

The authors provided thoughtful responses to the reviewers comments to improve the clarity of the manuscript and address concerns about the separation of local and non-local effects. I particularly appreciate the comparison of the local effects between the moving window regression and chessboard methods using the MPI-ESM (Fig A6). However, I still have concerns over the overestimation of the local cooling effects in the manuscript and the implications of such overestimation for the findings.

More specifically, in the limitations section of the Discussion, the authors mention that they did not find evidence of systematic underestimation of local effects (without scaling), but that scaling to 100% could lead to overestimation of local effects compared to other methods. When performing the chessboard calculation of local effects, I'm assuming a 100% deforestation was performed on each deforested grid cell when deforesting 1 out of 4 or 2 out of 4 grid cells (Fig. A6). Given that the deforest\_glob experiment only does partial deforestation in most grid cells, the deforest\_glob scaled to 100% appears to be the best comparison with the chessboard experiment. I am questioning whether the comparison without scaling is relevant (in this case, for a given deforested grid cell, you would be comparing local effects resulting from partial deforestation from the deforest\_glob experiment with local effects resulting from complete deforestation in the chessboard experiment)? If local effects were comparable across methods, shouldn't we see stronger local effects in the chessboard experiment in this case? When comparing the scaled deforest glob experiment and the chessboard experiment, there is evidence of overestimation of the local effects as pointed out by the authors. The authors acknowledge factors that could contribute to the overestimation of local effects due to scaling, but isn't it possible that the choice of methodology itself (moving window regression vs chessboard) could lead to an overestimation of local effects? I don't think we can say that it is only scaling that leads to the overestimation of local effects.

We are thankful for the reviewer's constructive comments, and apologize that some of our responses in the first Revision (R1) were perhaps not clear.

In regards to evaluating possible uncertainties due to choice of method for separating local and non-local biogeophysical effects due to deforestation in the ESMs, we compared our results (based on a modified moving window regression) to the findings from Winckler et a. (2019) (based on chessboard experiments) in Revision 1. We agree with the reviewer's comment that for this comparison at grid cell level, scaling to 100% is the most appropriate way and have revised Fig. A6 accordingly (see revised ms). The comparison to the chessboard pattern simulation with 1 out of 4 and 2 out of 4 grid cells being deforested was motivated by the similarity in the degree and spatial extent of deforestation in these simulations and *deforest-glob* (see Fig. R1)

Fig. R1: Comparison of degree of deforestation in chessboard pattern simulations from Winckler et al. (2019) and *deforest-glob*.

The comparison between local temperature sensitivities due to deforestation based on the two different separation methods do show some deviations including in *deforest-glob* a stronger cooling over large portions of Northern Eurasia and a stronger warming over central portions of the Amazon and African rainforests (Fig. A6 in ms). But the *deforest-glob* simulation also shows weaker cooling over parts of boreal North America and similarly weaker warming over northeast Brazil, southeast Africa and southeastern Asia (Fig. A6 in ms). Thus, while these results underscore uncertainties when separating local and non-local effects, they provide no evidence for systematic over- or underestimation of the local effects with our applied moving window regression separation method. To reflect on these results, we made the following changes in the revised ms in the Limitations section (Section 5):

"An additional analysis comparing temperature sensitivities to deforestation based on the linear regression method applied in this study and the chessboard pattern deforestation experiments of Winckler et al. (2019a) did reveal some differences in corresponding patterns but did not provide evidence of systematic under- or overestimation of local effects (Fig. A6)."

We would like to point out that *deforest-glob* and the chessboard experiments in Winckler et al. (2019) are different in many aspects (e.g. degree of deforestation, deforestation pattern), and a comparison of the results has, therefore, limited utility. To reflect on these aspects more clearly, we included under Limitations the following: "However, comparisons of the local biogeophysical effects among different deforestation scenarios – even within the same model framework - are challenging as local effects are influenced by the degree of deforestation (e.g., partial or complete), and the initial forest cover (Li et al., 2016; Winckler et al., 2017)."

Furthermore, the implications of such overestimation of local effects (regardless of where it comes from) for the findings are not discussed. How would that influence the emergent constraints discussed previously? Could this result in any changes in the relationships between the variables considered (local temperature, total temperature, albedo, latent heat flux)? For example, if the local effects are overestimated, the consistent overestimation of the local cooling response of the models in the boreal and temperate regions compared to observations may not be as pronounced as suggested in this study. Would that also explain part of the local albedo sensitivity which can't be explained by the higher percentage of snow cover in ESMs? Similarly, the authors also mention that the total cooling resulting from deforestation is overestimated in ESMs due to overestimation of the local cooling by ESMs (compared to observations), but that claim may also change if local effects are overestimated in this study. For example, if local effects are overestimated, it is possible that non-local effects are underestimated (by a similar amount

since a proportion of the cooling is just moved from local to non-local) which would then impact the relationship between local and total effects. Based on the evidence provided, particularly the stronger snow coverage in ESMs compared to observations, local (and total) cooling in northern latitudes do appear to be overestimated in ESMs compared to observations, but it may not be as pronounced as the results in this study are showing. Adding a discussion of the implications of an overestimation of local effects as part of the limitations would be important.

We agree with the reviewer, that a discussion on the implications of overestimation of local effects has not been included adequately. In the case of an overestimation of the local effect due to applying linear scaling (as done here), the validity of the emergent constraints – i.e., the linear relationship between local and total temperature sensitivities – is not affected as the same scaling approach is applied to each of the ESMs. If scaling is applied to observations as well (as in the case of MODIS observations) than the proximity between the ESM-based local effects and observational constraints would shift. We illustrated this in Fig. R2.

If a systematic overestimation or underestimation of the local effect is caused by the method used for separating local and non-local effects in ESMs, the inter-model comparison and emergent constraint would be still valid but the proximity to observed constraints would be affected. For example, if the local ESM temperature sensitivities in the boreal regions would be systematically overestimated or underestimated, their proximity to observed local temperature sensitivities would diminish or become larger (after bias correction), respectively. This is because the ESM based local sensitivities show enhanced cooling (see Fig. R2). Yet, we find no evidence of systematic biases of the local temperature sensitivities in ESMs based on our choice of separation method (see above).

Fig. R2: Hypothetical effect of overestimation of the local temperature sensitivity in the boreal region due to scaling (see Fig. 8a in ms). The observed constraints are highlighted in orange (Bright et al., 2017) and blue (MODIS).

To reflect on those points, we have added the following paragraph under Limitations: "Importantly, potential uncertainties associated with scaling or method of separating local and non-local effects do not alter the relationship between ESM-based total and local temperature sensitivities (Fig. 8 in ms) - as all models experience the same 'bias' - and therefore the slope of the linear relationship central to the emergent constraint concept is not affected. However, a

possible under- or overestimation of the ESM-based local effects could lead to shifts in the proximity to observational constraints."

I also have a few additional specific comments, mainly to enhance clarity.

Line 15: Consider specifying the linear relationships between what variables in that sentence (i.e. resulting from the linear relationships between X and Y within the model ensemble).

Changed to "(i.e. resulting from the linear relationships between local albedo and surface temperature within the model ensemble)"

Lines 89-106: Those are two important paragraphs setting the stage for the study. The stated goals of the study do not mention looking at the emerging relationship between local and total effect despite this being an important part of the manuscript. The stated goals focus mainly on the relationship between local temperature and biogeophysical properties within observations and models despite mentioning that knowing the emerging relationship between local and total effects within models would be useful. There seems to be a disconnect there.

We highlight the emerging relationship between local and total effects in the end of the first paragraph: "By knowing the local effects from observations and the emerging relationship between local and total effects from model experiments, we could use these relationships to constrain the total effects of deforestation on the near-surface climate."

To put more emphasize on this aspect, we have changed lines 103-105 as follow: "The objective is to provide observationally based emergent constraints for local surface temperature, albedo and latent heat flux sensitivities to deforestation, against which both the local and total (local and non-local) ESM based sensitivities can be compared."

**Line 178: extra comma**

**Comma removed.**

Line 230-232: That sentence is not clear, maybe it should be rephrased. It also does not flow well with the end of the last paragraph; it is missing a transition. It is not clear how the local surface temperature sensitivities are related to the relationship between local and total temperature in this sentence.

Lines 230-232 rephrased as follow: "By showing the linear relationship between local and total surface temperature change due to deforestation, we are able to constrain the overall response of ESMs to deforestation, for which no observations are available, as the range of plausible local surface temperature sensitivities is narrowed down by observations and total surface temperature responses are related to those by statistically and physically meaningful relationship.". This paragraph is moved up to improve the flow of the text.

Line 330: The link between this paragraph and the previous one where we talk about the relationship between local and total effect is not clear. Consider providing a better transition and elaborating on how Fig 9 ties to Fig 8.

Added at line 333: "As the local cooling effect of deforestation is strongest in the colder months, it is important to consider whether the overestimation can be observed also during the warmer months, when the albedo effects are not that pronounced."

Added at line 339: "confirming that the overestimation of cooling (Fig. 8) persists also during summer."

Line 359: Would that still be the case if local effects are overestimated in this study, as per the main comment above?

The statement will still be true because the spread between the models (which is independent of a potential overestimation bias as discussed above) is too large, so it is not possible that all ESMs fall within the boundaries of the observational constraints.

Line 487: overestimation of local and total effects, not local and non-local effects. The results did not discuss overestimation of non-local effects.

Fixed in revised manuscript.

Fig 5: FLUXNET-based surface temperature sensitivity should be green to match the green line of FLUXNET in Fig 4. Keep orange for MODIS.

Updated in revised manuscript.

Fig 6: Same comment as Fig 5. Also, why is Modis showing a negative local temperature change for all levels of snow cover, whereas it is positive in Fig 5 for the boreal region?

In Fig. 5 we show the annual temperature effect, while in Fig. 6 only March and April are considered, hence the difference in the MODIS-based temperature. The figure is updated in the revised manuscript.

Fig 7: Same comment as Fig 5.

Updated in revised manuscript.

Fig 8: To be consistent with the colors chosen in other figures, MODIS sensitivity uncertainty range should be orange and FLUXNET sensitivity uncertainty range should be green.

Updated in revised manuscript.

Fig A4: Same comment as Fig 5.

Updated in revised manuscript.

Fig A6: pattern, typo in the caption. See also main comment regarding comparison between deforest\_glob (not scaled) and chessboard experiments.

Typo fixed.

**Literature**

Li, Y., De Noblet-Ducoudré, N., Davin, E. L., Motesharrei, S., Zeng, N., Li, S., & Kalnay, E. (2016). The role of spatial scale and background climate in the latitudinal temperature response to deforestation. *Earth System Dynamics*, 7(1), 167-181.

Winckler, J., Reick, C. H., & Pongratz, J. (2017). Why does the locally induced temperature response to land cover change differ across scenarios?. *Geophysical Research Letters*, *44*(8), 3833-3840.